# Efficient Molecular Conformer Generation with SO(3)-Averaged Flow Matching and Reflow

Zhonglin Cao [* 1]  Mario Geiger [* 1]  Allan dos Santos Costa [1 2 3]  Danny Reidenbach [1]  Karsten Kreis [1]
Tomas Geffner [1]  Franco Pellegrini [1]  Guoqing Zhou [1]  Emine Kucukbenli [1]

## Abstract

Fast and accurate generation of molecular conformers is desired for downstream computational chemistry and drug discovery tasks. Currently, training and sampling state-of-the-art diffusion or flow-based models for conformer generation require significant computational resources. In this work, we build upon flow-matching and propose two mechanisms for accelerating training and inference of generative models for 3D molecular conformer generation. For fast training, we introduce the SO(3)-*Averaged Flow* training objective, which leads to faster convergence to better generation quality compared to conditional optimal transport flow or Kabsch-aligned flow. We demonstrate that models trained using SO(3)-*Averaged Flow* can reach state-of-the-art conformer generation quality. For fast inference, we show that the reflow and distillation methods of flow-based models enable few-steps or even one-step molecular conformer generation with high quality. The training techniques proposed in this work show a path towards highly efficient molecular conformer generation with flow-based models.

## 1. Introduction

Molecular conformer generation is the task to predict the ensemble of 3D conformations of molecules given their 2D molecular graphs (Hawkins, 2017). Generating high quality molecular conformers that fit their natural 3D structures is a crucial task for computational chemistry because many physical and chemical properties (Guimarães et al., 2012; Schwab, 2010; Shim & MacKerell Jr, 2011) are de-

termined by the conformers. In the domain of drug discovery, molecular conformer generation is a prerequisite for both structure-based (Trott & Olson, 2010) and ligand-based (Rush et al., 2005) compound virtual screening. For established computational chemistry molecular conformer generation tools, there is a trade-off between generation speed and the quality or diversity of generated conformers (Axelrod & Gomez-Bombarelli, 2022). For example, enhanced molecular dynamics simulation (Grimme, 2019) can generate diverse conformers by sampling the conformation space rather exhaustively, but this approach is slow due to the need of multiple energy function evaluations. RDKit (Landrum, 2016) and some rule-based tools (Hawkins et al., 2010) are faster but may miss many low-energy conformer, and the generation quality can deteriorate when molecule size increases. Thus, deep generative models are being sought as potential solutions to overcome such trade-off and bring fast, diverse, and high-quality molecular conformer generation.

Many earlier works are based on generative models (Simm & Hernández-Lobato, 2019; Zhu et al., 2022; Luo et al., 2021; Shi et al., 2021; Xu et al., 2022), given the stochastic nature of the molecular conformer generation task. However, established cheminformatics tools such as OMEGA (Hawkins et al., 2010) still have better generation quality and faster sampling speed compared to early deep-learning based methods. Torsional diffusion (Jing et al., 2022) is the first diffusion model to achieve better generation quality than cheminformatics model. By restricting the degree-of-freedom on the torsion angles, torsional diffusion can generate diverse conformers with a lightweight model and fewer number of reverse diffusion steps. More recent works, such as Molecular conformer field (MCF) (Wang et al., 2024) and ET-Flow (Hassan et al., 2024), perform diffusion or flow-matching directly on the Cartesian coordinates of the atoms. With more scalable transformer architectures, they have achieved the state-of-the-art conformer generation quality. However, iterative ODE or SDE solving with large transformer models to generate every conformer can still be computationally infeasible when the virtual screening library contains billions of compounds (Bellmann et al., 2022).

---

[*]Equal contribution  [1]NVIDIA [2]MIT Center for Bits and Atoms [3]Work was completed during internship with NVIDIA. Correspondence to: Zhonglin Cao <zhonglinc@nvidia.com>, Mario Geiger <mgeiger@nvidia.com>.

*Proceedings of the 42nd International Conference on Machine Learning*, Vancouver, Canada. PMLR 267, 2025. Copyright 2025 by the author(s).

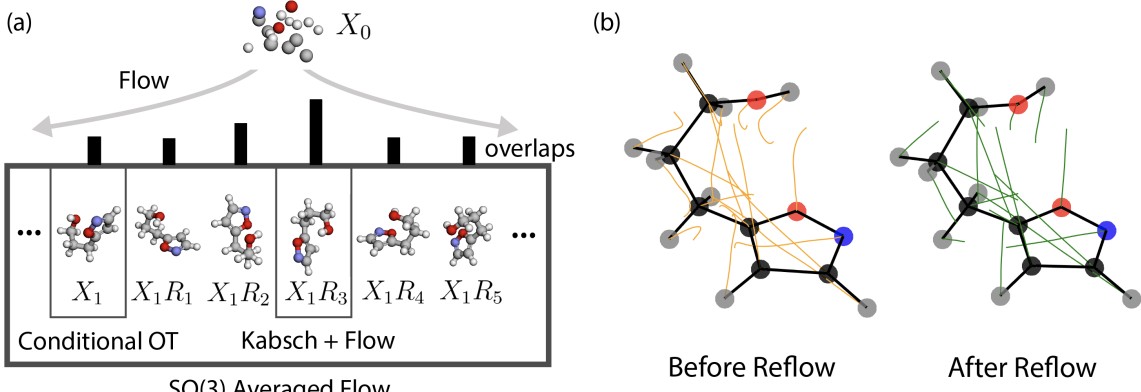

*Figure 1.* **SO(3)-Averaged Flow and Reflow (a)** We illustrate a comparison between our approach *Averaged Flow*, conditional OT and Kabsch + Flow. While conditional OT randomly assigns any rotation of the data, Kabsch + Flow assigns the rotation of largest overlap. Our method instead computes the expected flow across all rotations. **(b)** Flow trajectory visualization before and after the reflow with 100 Euler steps. The flow trajectories are effectively straightened after reflow. Carbon, hydrogen, oxygen, and nitrogen are colored as black, gray, red, and blue, respectively.

In this work, we propose a novel flow-matching training approach to improve the efficiency of deep learning model training and sampling for molecular conformer generation. To improve training efficiency, we design a new flow-matching objective called SO(3)-*Averaged Flow* (Fig. 1a). As an objective, *Averaged Flow* avoids the need to rotationally align prior and data distributions by *analytically* computing the averaged probability flow path from the prior to all the rotations of the data sample. Models trained with *Averaged Flow* are experimentally shown to converge faster to better performance. To improve the sampling efficiency, we adopt the *reflow* and distillation technique (Liu et al., 2022) to straighten the flow trajectories (Fig. 1b). Straightened trajectories enable high quality molecular conformer generation with few-step or even one-step ODE solving, thus significantly reducing the computational cost.

Our main contribution can be summarized as follows: **(i)** We propose a novel SO(3)-*Averaged Flow* training objective. *Averaged Flow* eliminates the need for rotational alignment between prior and data by training the model to learn the average probability path over all rotations of the data. This leads to faster convergence to better performance for molecular conformer generation, and can be extended to other similar tasks. Models trained using *Averaged Flow* can achieve state-of-the-art conformer generation quality. **(ii)** We adopt reflow and distillation to reduce the number of ODE steps required for the model to generate high quality conformers. Such technique significantly improves the sampling efficiency of flow-matching models in molecular conformer generation. **(iii)** Both *Averaged Flow* and reflow+distillation are *model architecture-agnostic*, meaning that our proposed methods can be directly applicable to both equivariant and non-equivariant neural network architectures.

## 2. Background and Related Works

### 2.1. Generative Models for Conformer Generation

The task of molecular conformer generation in its core is to sample from the intractable conformer distribution conditioned on the 2D molecular graph. Therefore, generative deep learning models are well-suited for this task, and many methods have been proposed. Deep learning model are usually trained on datasets containing molecular conformers generated by CREST (Pracht et al., 2020), which uses computationally expensive semi-empirical quantum chemistry method (Bannwarth et al., 2019) under the hood. The earliest works in this field use variational autoencoder to generate intrinsic inter-atomic distance (Simm & Hernández-Lobato, 2019; Xu et al., 2021). Shi et al. (2021) proposes a score-matching method that learns the gradient of intrinsic atom coordinates in molecular graph. Ganea et al. (2021) addresses molecular conformer generation by designing a message passing neural network to predict the local 3D structure and torsion angles. Xu et al. (2022) adopts a diffusion model and equivariant graph neural network to generate molecular conformers by iteratively denoising the Euclidean atom coordinates from sampled noise. Besides generating conformers from scratch, some works focus on optimizing molecular conformers to lower energy states (Lee et al., 2024a; Guan et al., 2021). Torsional diffusion (Jing et al., 2022) reduces the degree-of-freedom by refining the torsion angles of RDKit-generated (Landrum, 2016) initial conformers with a diffusion process on the hypertorus. Such design allows torsional diffusion to significantly reduce sampling steps. One drawback of torsional diffusion is that it relies on an RDKit-generated conformer as the starting point of diffusion, which adds computational overhead to the generation process. The generation quality of RDKit, especially for atom coordinates in rings, can also impact the sample

quality of torsional diffusion. Molecular conformer field (MCF) proposed by Wang et al. (2024) is a recent work that leverages the scaling power of the transformer architecture (Jaegle et al., 2021) and diffusion model. MCF achieves state-of-the-art performance in molecular conformer generation by training models with tens to hundreds million of parameters to denoise the atoms' Euclidean coordinates using DDPM paradigm (Ho et al., 2020). Equivariant Transformer Flow (ET-Flow) is a concurrent work that trains an equivariant flow-matching model to generate conformers from a prior distribution. By combining harmonic prior (Jing et al., 2023), flow-matching, and Kabsch alignment that reduces transport cost, ET-Flow is reported to outperform MCF on several metrics with fewer ODE steps.

Overall, the trade-off between conformer generation quality and speed is a prevailing issue. Specifically, semi-empirical quantum chemistry can sample very high-quality conformers at high computational cost. Diffusion or flow-matching models can generate high-quality conformers but the iterative ODE/SDE solving process can be slow, making them less practical for large-scale virtual screening. Cheminformatics tools such as RDKit and OMEGA are very fast but generate conformers with limited diversity.

## 2.2. Flow-Matching

*Averaged Flow* is based on Flow Matching (Lipman et al., 2023; Liu et al., 2023a; Albergo & Vanden-Eijnden, 2023), which models a probability density path $p_t(\mathbf{x}_t)$ that gradually transforms an analytically tractable noise distribution ($t = 0$) into a data distribution ($t = 1$), following a time variable $t \in [0, 1]$. Formally, the path $p_t(\mathbf{x}_t)$ corresponds to a *flow* $\psi_t$ that pushes samples from $p_0$ to $p_t$ via $p_t = [\psi]_t * p_0$, where $*$ denotes the push-forward. In practice, the flow is modeled via an ordinary differential equation (ODE) $dx_t = v_t^\theta(x_t)dt$, defined through a learnable vector field $v_t^\theta(x_t)$ with parameters $\theta$. Initialized from noise $x_0 \sim p_0(x_0)$, this ODE simulates the flow and transforms noise into approximate data distribution samples. The probability density path $p_t(x_t)$ and the (intractable) ground-truth vector field $u_t(x_t)$ are related via the continuity equation $dp_t(x)/dt = -\nabla_x \cdot (p_t(x)u_t(x))$. To construct $p_t$, Lipman et al. (2023) introduce a conditional probability $p_t(x|x_1)$ and conditional vector field $u_t(x|x_1)$, both related to their unconditional counterparts as follow:

$$p_t(x) = \int p_t(x|x_1)q(x_1)dx_1. \tag{FM6}$$

$$u_t(x) = \int u_t(x|x_1)\frac{p_t(x|x_1)q(x_1)}{p_t(x)}dx_1 \tag{FM8}$$

With the following simple choices of conditional probability and flow

$$p_t(x|x_1) = \mathcal{N}(x; \mu_t(x_1), \sigma_t^2(x_1)) \tag{FM10}$$

$$\psi_t(x) = \sigma_t(x_1)x + \mu_t(x_1) \tag{FM11}$$

they prove that

$$u_t(x|x_1) = \frac{\sigma_t'(x_1)}{\sigma_t(x_1)}(x - \mu_t(x_1)) + \mu_t'(x_1). \tag{FM15}$$

It is noteworthy that we refer to the linear interpolant $x_t = tx_1 - (1 - t)x_0$ between the noise and data distribution as conditional optimal transport (OT) following Lipman et al. (2023).

## 2.3. Techniques to Improve Sampling Efficiency

With the success of denoising diffusion probabilistic models (Ho et al., 2020), much attention has been drawn to improve the sampling speed of diffusion models. DDIM (Song et al., 2020) shows that the sampling steps can be significantly reduced by formulating the sampling process as ODE solving. Knowledge distillation techniques (Meng et al., 2023; Salimans & Ho, 2022; Song et al., 2023; Song & Dhariwal, 2023) have also been proposed to reduce sampling steps and accelerate generation. Rectified flow (Liu et al., 2022; Liu, 2022) is a method proposed to train the model to learn straight probability flow that bridges prior and data distribution. The *reflow* technique in rectified flow can straighten the flow trajectory and reduce the transport cost, allowing generation in very few steps with high quality. After reflow, the model can be further distilled to improve one-step generation. The reflow and distillation technique has been proven effective in enabling few-step or even single-step text-to-image (Esser et al., 2024; Liu et al., 2023b) and point cloud (Wu et al., 2023) generation.

## 3. Method

### 3.1. SO(3)-*Averaged Flow*

The concept of *Averaged Flow* involves recognizing that the data distribution $q$ may exhibit group symmetries, which can be explicitly integrated out. A symmetry group $G$ of $q$ consists of transformations $g : x \mapsto g \cdot x$ that leave the distribution $q$ unchanged, meaning $q(x) = q(g \cdot x)$.

If we focus on Lie groups with a Haar measure (Zee, 2016; Nachbin & Bechtolsheim, 1965; Chirikjian & Kyatkin, 2000), we can express $q$ as

$$q(x) = \int d\hat{x}\, \hat{q}(\hat{x}) \int dg\, \delta_{g \cdot \hat{x}}(x) \tag{1}$$

where $\hat{q}$ represents the distribution over the group orbits, $\hat{x}$ is a representative point of the orbit, and the integral over $G$ uses the Haar measure. By substituting this into equation FM8, we obtain the vector field:

$$u_t(x) = \int d\hat{x}\, \hat{q}(\hat{x}) \int dg\, u_t(x|g \cdot \hat{x})\frac{p_t(x|g \cdot \hat{x})}{p_t(x)} \tag{2}$$

Notice that $p_t(x) = \int d\hat{x}\, \hat{q}(\hat{x}) \int dg\, p_t(x|g \cdot \hat{x})$ is the partition function.

Let's consider the case of conformer generation:

1. $x$ is a $N \times 3$ matrix representing the 3D coordinates of $N$ atoms.

2. The group $G$ is the rotation group $SO(3)$. We will use $R$ to denote the rotation matrix, which acts on $x$ as $x \mapsto xR^T$.

3. The goal is to generate molecular conformers that correspond to at least local minima in the conformational energy landscape. The orbits $\hat{x}$ in this case correspond to the different low-energy conformers of a given molecule and their permutations that leave the 2D molecular graph invariant. Therefore, the integral $\int d\hat{x}\, \hat{q}(\hat{x})$ in Eq.2, representing the entire conformer ensemble, can be written as $\sum_{\hat{x} \in \mathcal{X}} \hat{q}(\hat{x})$, where $\mathcal{X}$ is the set of conformers and $\hat{q}(\hat{x})$ is the weight associated with each conformer.

4. $p_t(x|x_1)$ is a Gaussian of the form:

$p_t(x|x_1)$

$\propto \exp\left( \frac{1}{2} \frac{1}{(1-t)^2} \sum_{ij\delta} (x - tx_1)_{i\delta} \Sigma_{ij} (x - tx_1)_{j\delta} \right)$

$\equiv \exp\left( \frac{1}{2} \frac{\|x - tx_1\|_\Sigma^2}{(1-t)^2} \right)$

where $\Sigma$ is a $\mathbb{R}^{N \times N}$ matrix. We will use the notation $\|A\|_\Sigma^2 = \text{tr}(A^T \Sigma A)$.

We can rewrite the vector field $u_t(x)$, averaged over all conformers and the $SO(3)$ group, in this case as:

$$u_t(x) =$$

$$\frac{1}{Z_t(x,0)} \sum_{\hat{x} \in \mathcal{X}} \hat{q}(\hat{x}) \int_{SO(3)} dR\, \frac{\hat{x}R^T - x}{1-t} e^{-\frac{1}{2}\frac{\|x - t\hat{x}R^T\|_\Sigma^2}{(1-t)^2}}$$

$$(3)$$

and define $Z_t(x, \alpha)$ as:

$$Z_t(x, \alpha) = \sum_{\hat{x} \in \mathcal{X}} \hat{q}(\hat{x}) \int_{SO(3)} dR\, e^{-\frac{1}{2}\frac{\|x - t\hat{x}R^T\|_\Sigma^2}{(1-t)^2} + \text{tr}(\alpha^T \hat{x}R^T)}$$

$$(4)$$

where $\alpha$ is an $N \times 3$ matrix that will be needed in the following steps.

Note $u_t(x)$ can be calculated as the derivative of $\log Z_t(x, \alpha)$ with respect to $\alpha$, evaluated at $\alpha = [0]_{N \times 3}$,

$$u_t(x_t) = ([\partial_\alpha \log Z_t(x_t, \alpha)]_{\alpha=[0]_{N \times 3}} - x_t)/(1-t). \quad (5)$$

The integral over $R$ can be computed using the formula from Mohlin et al. (2020), which provides a closed-form

solution for

$$F \mapsto \log \int_{SO(3)} dR \exp(\text{tr}(FR^T)) \quad (6)$$

where $F$ can be any $3 \times 3$ matrix. We can now leverage Eq. 6 to solve for $\log Z_t(x_t, \alpha)$. Expanding the quadratic term in Eq. 4, $-\frac{1}{2}\frac{\|x - t\hat{x}R^T\|_\Sigma^2}{(1-t)^2} = -\frac{1}{2}\frac{\text{tr}(x^T\Sigma x) + t^2\text{tr}(\hat{x}^T\Sigma\hat{x}R^TR) - 2t\,\text{tr}(x^T\Sigma\hat{x}R^T)}{(1-t)^2}$, we obtain the following equation:

$$\log Z_t(x, \alpha) =$$

$$\log \sum_{\hat{x} \in \mathcal{X}} \hat{q}(\hat{x}) \exp\left( \underbrace{\log \int_{SO(3)} dR\, e^{\text{tr}((\alpha^T + \frac{t}{(1-t)^2}x^T\Sigma)\hat{x}R^T)}}_{\text{closed-form solution using } F = \alpha^T\hat{x} + \frac{t}{(1-t)^2}x^T\Sigma\hat{x}} - c \right)$$

$$(7)$$

where $c = \frac{\text{tr}(x^T\Sigma x) + t^2\text{tr}(\hat{x}^T\Sigma\hat{x})}{2(1-t)^2}$. Since we will take the derivative with respect to $\alpha$ to compute $u_t(x)$, all terms that neither depend on $\alpha$ nor $R$ (note that $R^TR$ is the identity) will contribute as multiplicative factors to the integral. Now, we can plug the *analytically* solved $\partial_\alpha \log Z_t(x_t, \alpha)$ into Eq. 5 to directly learn the $SO(3)$-*Averaged Flow* $u_t(x_t)$ with:

$$\mathcal{L}_{\text{AvgFlow}}(\theta) = \mathbb{E}\left[ \|v_t^\theta(x_t) - u_t(x_t)\|^2 \right], \text{with } t \in [0, 1]. \quad (8)$$

We provide the Python implementation of this formula in Appendix C.1. While our *Averaged Flow* implementation is capable of handling multiple conformer states in the summation in Eq 7, in practice, we approximate the expectation of the conformer ensemble by sampling one conformer in each training epoch. Following previous works (Jing et al., 2022), $\hat{q}(\hat{x})$ is taken as a uniform distribution over all conformers. The computation time benchmark (Table C.2) shows that only a small overhead is introduced when using the *Averaged Flow* objective. We empirically find that the choice of interpolant ($x_t$) during training should depend on the model architecture. If equivariant neural network is used as $v_t^\theta$, a *linear interpolant* (Liu et al., 2022; Lipman et al., 2023)

$$x_t = t \cdot x_0 + (1-t) \cdot x_1 \quad (9)$$

can be used. However, if a non-equivariant neural network is used as $v_t^\theta$, training with *Averaged Flow* requires the interpolant to be solved with simulating the probability flow ODE:

$$x_t = x_0 + \int_0^t u_\tau(x_\tau)\mathrm{d}\tau \quad (10)$$

where $u_\tau(\cdot)$ is the ground-truth *Averaged Flow* function (Eq. 5). In practice, we simulate the ODE with fixed 20 Euler steps to compute $x_t$, which we refer to as the *integration interpolant*.

## 3.2. Reflow and Distillation

Diffusion and flow-based molecular conformer generation models typically require hundreds or even thousands of steps of numerical ODE or SDE solving during the sampling process. Such iterative processes add computational overhead and hinder the adoption of these model in industrial-scale downstream applications, which require fast generation. One effective technique to reduce the sampling steps without significantly sacrificing generation quality is to straighten the trajectory. Inspired by the success of such technique in point-cloud generation (Wu et al., 2023) and text-to-image generation (Esser et al., 2024; Liu et al., 2023b), we finetune our model $v_t^\theta$, trained with Averaged Flow, using the *reflow* algorithm proposed in previous rectified flow works (Liu et al., 2022; Liu, 2022). Specifically, we first randomly sample atom coordinates $x_0'$ from standard Gaussian and generates the corresponding conformer $x_1'$ by simulating the learned flow with ODE. The coupling $(x_0', x_1')$ is then used in the rectified flow objective to finetune the model:

$$\mathcal{L}_{\text{Reflow}}(\theta) = \mathbb{E}\Big[\|v_t^\theta(x_t', t) - (x_1' - x_0')\|^2\Big], \text{with } t \in [0, 1] \tag{11}$$

Liu et al. (2022) proved that the coupling $(x_0', x_1')$ yields equal or lower transport cost than $(x_0, x_1)$, where $x_0$ is sampled from the noise distribution and $x_1$ from the data distribution. Therefore, applying the reflow algorithm to fine-tune model with Eq. 11 can effectively reduce the transport cost and straighten the trajectory.

We empirically find that the transport trajectories bridging Gaussian noise and molecular conformers demonstrates high curvature when $t$ is close to 0 (one example shown in Fig. 1b). Therefore, inspired by Lee et al. (2024b), we sample $t$ from an exponential distribution with the probability density function as $p(t) \propto \text{Exp}(\lambda t)$, where $\lambda = -1.2$ by selection to focus the training more on $t < 0.5$. The distribution of $t$ is visualized in Fig. 7. After reflow, the sampling speed can be further reduced by distilling the relationship of the coupling $(x_0', x_1')$ into model $v_\theta$ to enable one-step transport and eliminate the need of ODE solving. During the distillation stage, we fine-tune the reflowed model $v_\theta$ with the following loss function:

$$\mathcal{L}_{\text{Distill}}(\theta) = \mathbb{E}\Big[\|v_t^\theta(x_0', 0) - (x_1' - x_0')\|^2\Big] \tag{12}$$

which is equivalent to Eq. 11 with $t = 0$.

## 3.3. Flow-Matching Model Architecture

To demonstrate that the *Averaged Flow* training objective (Eq. 8) is architecture-agnostic, we implemented and trained two different neural network for flow-matching including *(i)* a SE(3)-equivariant graph neural networks (NequIP) which is modified based from Batzner et al. (2022), and *(ii)* a non-equivariant yet highly scalable diffusion transformer

with pairwise bias (DiT). Both model takes in the featurized molecular graph as input. The featurization details can be found in Sec. B.2.

For the NequIP model (Fig.5), the features of each atoms ($\mathbf{Z}$) and bonds ($\mathbf{E}$) are first embedded by the model into scalar features, along with coordinate of atoms ($x_t$) at given timestep $t$. These features are then mixed with the edge vector through 6 interaction blocks of the model. Finally, a linear layer is used to make prediction of the vector field as type $l = 1$ geometric features. Noteworthy modifications to the original architecture include incorporating edge features into the graph convolution layer, adding residual connections, and equivariant layer normalization to stabilize training. Our modified NequIP model contains ∼4.7 million parameters and its details can be found in Sec.A.1.

---

**Algorithm 1** *Averaged Flow* with Reflow+Distillation Train

---

**Require:** Molecule Dataset $\mathcal{G} = [G_0, ..., G_D]$, each with conformers $\mathcal{X}^G = [x^{G,0}, ...x^{G,N}]$
**Require:** Learnable Velocity Field Network $v^\theta$
  **1. Base SO(3) Averaged Flow Training**
  $t, x_0, G \sim \mathcal{U}(0,1), \mathcal{N}(0,1), \mathcal{G}$
  $x_1 \sim \mathcal{X}^G$
  **if** $v_t^\theta$ is equivariant **then**
    $x_t \leftarrow t \cdot x_0 + (1 - t) \cdot x_1$ (*linear interpolant* Eq.9)
  **else**
    $x_t \leftarrow x_0 + \int_0^t u_\tau(x_\tau)\mathrm{d}\tau$ (*integration interpolant* Eq.10)
  **end if**
  $u_t(x_t) \leftarrow$ Solve closed-form Eq. 5 for $x_t$ and $t$
  Gradient Step $-\|v_t^\theta(x_t|G) - u_t(x_t)\|^2$
  **2. Reflow**
  $x_0' \sim \mathcal{N}(0,1)$
  $x_1' \sim \text{ODESolve}\big(v_t^\theta(\cdot|G), x_0'\big)$
  Finetune model with coupled pair $(x_0', x_1')$ through Eq. 11
  **3. Distillation**
  Train model with coupled pair $(x_0', x_1')$ through Eq. 12

---

Our DiT model (Fig. 6) is based on Diffusion Transformers (Peebles & Xie, 2023). To include the critical covalent bond information and add extra structural details, a pairwise representation is constructed using the pairwise distances between atoms and bond features, and is used to inject additional learnable bias in the attention mechanism. This design is inspired by AlphaFold3 (Abramson et al., 2024) and has proven highly effective in the protein generation task (Geffner et al., 2025). Our DiT model contains ∼52 million parameters. We also trained a slightly larger variant of it, DiT-L, which contains ∼64 million parameters to match the size of a baseline MCF (Wang et al., 2024). Details of our DiT models can be found in Sec.A.2. Both model are trained and fine-tuned using *Averaged Flow* + reflow + distillation following the Algorithm 1. Details of model training and sampling are included in Sec. B.3.

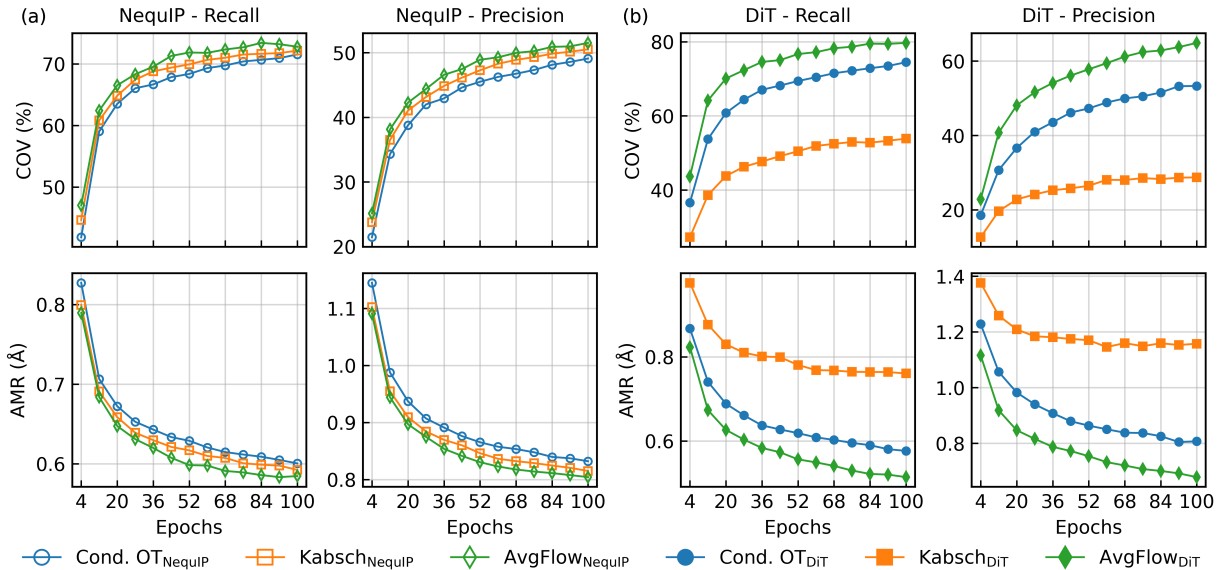

*Figure 2.* **Comparison between training objectives.** Both (a) NequIP and (b) DiT model trained with *Averaged Flow* consistently converge to better performance on a 300-molecule GEOM-Drugs test subset. The other two objectives we compared *Averaged Flow* to are: *(i)* Conditional OT and *(ii)* Kabsch alignment of noise $x_0$ with conformer $x_1$ before conditional OT (Hassan et al., 2024).

## 4. Experiments

Following previous works, we train and evaluate our model on the GEOM-QM9 and GEOM-Drugs datasets (Axelrod & Gomez-Bombarelli, 2022). We follow the splitting strategy proposed by Ganea et al. (2021); Jing et al. (2022), and test our model on the same test set containing 1000 molecules for both QM9 and Drugs datasets. Dataset and splitting details are included in Sec. B.1. The major model evaluation metrics are the average minimum RMSD (AMR, the lower the better) and coverage (COV, the higher the better). Both AMR and coverage are reported for precision (AMR-P and COV-P) and recall (AMR-R and COV-R). The definition of metrics are specified in Sec.B.6. Intuitively, coverage measures the percentage of ground truth conformers that are generated (recall) or the percentage of generated conformers that are close enough to ground truth (precision), while AMR measures the average RMSD between each ground truth and its closest generated conformer (recall), or vice versa (precision). There are three types of baselines in this work: *(a)* methods with fast inference speed such as cheminformatics tools (RDKit, OMEGA) and regression model GeoMol (Ganea et al., 2021); *(b)* lightweight diffusion models with reduced degree-of-freedom (Torsional Diffusion); and *(c)* large transformer-based diffusion or flow-based model operating on Euclidean atomistic coordinates (MCF and ET-Flow). It is worth mentioning that MCF has three variants, with number of parameters ranging from 13 to 242 millions. MCF uses DDPM sampler for full SDE simulation and DDIM sampler for few-step generation. ET-Flow has two variants: ET-Flow and ET-Flow-SS. The former is shown to produce better few-step generation quality, while the latter is shown to have better generation

quality with more simulation steps. Moreover, to fairly validate the effectiveness of the *Averaged Flow* objective, we compare the performance of both the NequIP and DiT architectures (architecture details in Appendix A) trained with different objectives on the Drugs dataset. Similarly, we compare the performance of the NequIP architecture before and after reflow+distillation to show the necessity of reflow for few-step generation.

### 4.1. *Averaged Flow* Leads to Faster Convergence to Better Performance

To showcase the advantage of the *Averaged Flow* over other training objectives, we evaluate the performance of models trained on different objectives using a randomly sampled GEOM-Drugs test subset containing 300 molecules. The two other objectives compared are conditional OT and Kabsch alignment. The Kabsch alignment objective is to rotationally align the sampled noise $x_0$ with conformer $x_1$ before training with the conditional OT objective. Models are evaluated every 8 epochs of training starting from 4 to 100 epochs. Fig. 2 demonstrates that both models trained with *Averaged Flow* are consistently better than those trained with conditional OT and Kabsch alignment on all four metrics.

With only 68 epochs of training, $AvgFlow_{\text{NequIP}}$ has COV-R higher and AMR-R lower than the same architecture trained with the other two objectives for 100 epochs (Fig.2a). The COV-P (49.3%) and AMR-P (0.831Å) of $AvgFlow_{\text{NequIP}}$ trained for 52 epochs are better than those of Cond. $OT_{\text{NequIP}}$ (COV-P = 49.1% and AMR-P = 0.832Å) trained for 100 epochs. Also, $AvgFlow_{\text{NequIP}}$ outperforms $Kabsch_{\text{NequIP}}$ trained for 100 epochs on

AMR-P ($AvgFlow_{\text{NequIP}}$ = 0.814Å and $Kabsch_{\text{NequIP}}$ = 0.815Å) and on COV-P ($AvgFlow_{\text{NequIP}}$ = 50.9% and $Kabsch_{\text{NequIP}}$ = 50.5%) after 76 and 84 epochs, respectively.

The benefit of *Averaged Flow* is more significant with the non-equivariant DiT architecture (Fig.2b). $AvgFlow_{\text{DiT}}$ trained for only 12 epochs has better performance on all metrics than $Kabsch_{\text{DiT}}$ trained for 100 epochs. Compared to Cond. $OT_{\text{DiT}}$ trained for 100 epochs, $AvgFlow_{\text{DiT}}$ achieves better performance on precision metrics after 36 epochs of training and on recall metrics after 52 epochs of training. Overall, models trained with *Averaged Flow* converge in fewer epochs to better performance in molecular conformer generation.

### 4.2. GEOM-QM9

*Table 1.* **GEOM-QM9 Benchmark.** Quality of generated conformer ensembles for GEOM-QM9 ($\delta = 0.5$Å) test set in terms of Coverage (COV) and Average Minimum RMSD (AMR). Best scores are **bold** and second best underlined. Baseline values are taken from the corresponding papers.

| Model | COV-R (%)↑ | | AMR-R (Å)↓ | | COV-P (%)↑ | | AMR-P (Å)↓ | |
|---|---|---|---|---|---|---|---|---|
| | Mean | Med | Mean | Med | Mean | Med | Mean | Med |
| *Full Simulation and Non-diffusion/flow Baselines* | | | | | | | | |
| RDKit | 85.1 | 100 | 0.235 | 0.199 | 86.8 | 100 | 0.232 | 0.205 |
| OMEGA | 85.5 | 100 | 0.177 | 0.126 | 82.9 | 100 | 0.224 | 0.186 |
| GeoMol | 91.5 | 100 | 0.225 | 0.193 | 87.6 | 100 | 0.27 | 0.241 |
| Tor. Diff. | 92.8 | 100 | 0.178 | 0.147 | 92.7 | 100 | 0.221 | 0.195 |
| ET-Flow-SS (8.3M) | 95.0 | 100 | 0.083 | 0.035 | 91.0 | 100 | 0.116 | 0.047 |
| MCF-B (64M) | 95.0 | 100 | 0.103 | 0.044 | 93.7 | 100 | 0.119 | 0.055 |
| $AvgFlow_{\text{NequIP}}$ (4.7M) | **96.4** | 100 | 0.089 | 0.042 | 92.8 | 100 | 0.132 | 0.084 |
| $AvgFlow_{\text{DiT}}$ (52M) | 96.0 | 100 | **0.082** | **0.030** | **95.0** | 100 | **0.088** | **0.039** |
| *Two-step Generation* | | | | | | | | |
| $AvgFlow_{\text{NequIP-R}}$ (4.7M) | 95.9 | 100 | 0.151 | 0.104 | 87.7 | 100 | 0.236 | 0.207 |
| *One-step Generation* | | | | | | | | |
| $AvgFlow_{\text{NequIP-D}}$ (4.7M) | 95.1 | 100 | 0.220 | 0.195 | 84.8 | 100 | 0.304 | 0.283 |

On the GEOM-QM9 dataset, we compare our model with two widely used cheminformatics tools: RDKit and OMEGA[1], along with GeoMol (Ganea et al., 2021), Torsional Diffusion (Jing et al., 2022), ET-Flow-SS (Hassan et al., 2024), and MCF (Wang et al., 2024). We denote our NequIP model trained with *Averaged Flow* as $AvgFlow_{\text{NequIP}}$ and the DiT model as $AvgFlow_{\text{DiT}}$. The model finetuned with reflow and distillation are denoted as $AvgFlow_{\text{architecture-R}}$ and $AvgFlow_{\text{architecture-D}}$, respectively. Table. 1 shows that $AvgFlow_{\text{NequIP}}$ outperforms all other models in the COV-R metrics and nearly matches the AMR-R of ET-Flow-SS, indicating it is capable of generating very diverse conformers on the GEOM-QM9 dataset. The $AvgFlow_{\text{DiT}}$ model establishes a new state-of-the-art on the GEOM-QM9 dataset by outperforming all baselines in all metrics. More importantly, the $AvgFlow_{\text{NequIP-R}}$ and $AvgFlow_{\text{NequIP-D}}$ achieve higher COV-R than other models with only 2-step and 1-step ODE sampling, respectively. $AvgFlow_{\text{NequIP-R}}$ also outperforms all cheminformatics tools and GeoMol in all metrics. The benchmark on GEOM-QM9 shows that our model can achieve state-of-

the-art conformer generation performance on smaller scale molecules. Table. 1 also shows that reflow and distillation can effectively maintain the conformer generation quality of flow-based model with only 1 or 2 steps of ODE solving.

### 4.3. GEOM-Drugs

We then train and benchmark our models on GEOM-Drugs, which is a larger dataset containing conformers of drug-like molecules. The top section of Table 2 shows a comparison between our models and all baselines with full simulation (no SDE/ODE step limit). $AvgFlow_{\text{NequIP}}$ demonstrates good performance on GEOM-Drugs by outperforming torsional diffusion on all metrics. Compared with MCF-S which has approximately 3 times more parameters, $AvgFlow_{\text{NequIP}}$ achieves better COV-P and AMR-P, indicating more conformers generated by $AvgFlow_{\text{NequIP}}$ are close to ground truth conformers. With a more scalable and expressive architecture, $AvgFlow_{\text{DiT}}$ achieves performance on par with both MCF and ET-Flow-SS in full simulation conformer generation. Specifically, it outperforms MCF in precision metrics and surpasses ET-Flow-SS in recall metrics. Moreover, when scaled up to the sames number of parameters as MCF-B (64M), $AvgFlow_{\text{DiT-L}}$ outperforms all MCF variants in precision metrics and MCF-B in AMR-R. $AvgFlow_{\text{DiT-L}}$ also outperforms ET-Flow-SS in all metrics except only for AMR-P.

In the lower two sections of Table 2, we demonstrate the two-step and one-step generation benchmarks of our models, MCF, and ET-Flow. $AvgFlow_{\text{NequIP-R}}$ (two-step) can outperform cheminformatics tools and GeoMol on all metrics, with a large margin specifically on the recall metrics. $AvgFlow_{\text{DiT-R}}$ outperforms all baselines in coverage metrics of two-step generation. Most notably, our $AvgFlow_{\text{DiT-D}}$ significantly outperforms all baselines by a wide margin in one-step generation, thanks to the straightened trajectory (Fig. 8). We want to emphasize that it surpasses 20 steps of Tor. Diff. with one-shot generation, despite Tor. Diff. starting generation with RDKit-generated conformers. Furthermore, $AvgFlow_{\text{DiT-D}}$ (one-step) outperforms MCF-S (1000 steps full SDE simulation) across all precision metrics and exceeds the performance of all MCF and ET-Flow (two-step) models in the coverage metrics. Overall, the training strategy combining *Averaged Flow* with reflow and distillation enables a scalable transformer-based architecture like DiT to achieve exceptional one-shot conformer generation quality and diversity. More visualizations of generated conformers and ODE trajectories before and after reflow and distillation are included in Fig. 8. The exceptional one-step generation quality of $AvgFlow_{\text{DiT-D}}$ pushes the limit of the quality-speed trade-off in molecular conformer generation, giving it the potential to be adopted for large-scale virtual screen use cases.

---

[1]Results adopted from (Jing et al., 2022)

*Table 2.* **GEOM-Drugs Benchmark.** Quality of generated conformer ensembles for GEOM-DRUGS ($\delta = 0.75$Å) test set in terms of Coverage (COV) and Average Minimum RMSD (AMR). Best scores are **bold** and second best underlined. Baseline values are taken from the corresponding papers. *Due to the use of adaptive step size, the number of steps of $AvgFlow_{\text{NequIP}}$ is an average value over all test set molecules.

| Method | Step | COV-R (%) ↑ | | AMR-R (Å) ↓ | | COV-P (%) ↑ | | AMR-P (Å) ↓ | |
| | | Mean | Med | Mean | Med | Mean | Med | Mean | Med |
|---|---|---|---|---|---|---|---|---|---|
| ***Full Simulation and Non-diffusion/flow Baselines*** | | | | | | | | | |
| RDKit | - | 38.4 | 28.6 | 1.058 | 1.002 | 40.9 | 30.8 | 0.995 | 0.895 |
| OMEGA | - | 53.4 | 54.6 | 0.841 | 0.762 | 40.5 | 33.3 | 0.946 | 0.854 |
| GeoMol | - | 44.6 | 41.4 | 0.875 | 0.834 | 43.0 | 36.4 | 0.928 | 0.841 |
| Tor. Diff. | 20 | 72.7 | 80.0 | 0.582 | 0.565 | 55.2 | 56.9 | 0.778 | 0.729 |
| ET-Flow-SS (8.3M) | 50 | 79.6 | 84.6 | 0.439 | 0.406 | 75.2 | 81.7 | 0.517 | **0.442** |
| MCF-S (13M) | 1000 | 79.4 | 87.5 | 0.512 | 0.492 | 57.4 | 57.6 | 0.761 | 0.715 |
| MCF-B (64M) | 1000 | 84.0 | 91.5 | 0.427 | 0.402 | 64.0 | 66.2 | 0.667 | 0.605 |
| MCF-L (242M) | 1000 | **84.7** | **92.2** | **0.390** | **0.247** | 66.8 | 71.3 | 0.618 | 0.530 |
| $AvgFlow_{\text{NequIP}}$ (4.7M) | 102* | 76.8 | 83.6 | 0.523 | 0.511 | 60.6 | 63.5 | 0.706 | 0.670 |
| $AvgFlow_{\text{DiT}}$ (52M) | 100 | 82.0 | 86.7 | 0.428 | 0.401 | 72.9 | 78.4 | 0.566 | 0.506 |
| $AvgFlow_{\text{DiT-L}}$ (64M) | 100 | 82.0 | 87.3 | 0.409 | 0.381 | **75.7** | **81.9** | **0.516** | 0.456 |
| ***Two-step Generation*** | | | | | | | | | |
| MCF-B (64M) | 2 | 46.7 | 42.4 | 0.790 | 0.791 | 21.5 | 13.2 | 1.155 | 1.160 |
| MCF-L (242M) | 2 | 54.2 | 54.4 | 0.752 | 0.746 | 25.7 | 18.8 | 1.119 | 1.115 |
| ET-Flow (8.3M) | 2 | 73.2 | 76.6 | 0.577 | 0.563 | **63.8** | **67.9** | **0.681** | **0.643** |
| $AvgFlow_{\text{NequIP-R}}$ (4.7M) | 2 | 64.2 | 67.7 | 0.663 | 0.661 | 43.1 | 38.9 | 0.871 | 0.853 |
| $AvgFlow_{\text{DiT-R}}$ (52M) | 2 | **75.7** | **81.8** | **0.545** | **0.533** | 57.2 | 59.0 | 0.748 | 0.705 |
| ***One-step Generation*** | | | | | | | | | |
| MCF-B (64M) | 1 | 22.1 | 6.9 | 0.962 | 0.967 | 7.6 | 1.5 | 1.535 | 1.541 |
| MCF-L (242M) | 1 | 27.2 | 13.6 | 0.932 | 0.928 | 8.9 | 2.9 | 1.511 | 1.514 |
| ET-Flow (8.3M) | 1 | 27.6 | 8.8 | 0.996 | 1.006 | 25.7 | 5.8 | 0.939 | 0.929 |
| $AvgFlow_{\text{NequIP-D}}$ (4.7M) | 1 | 55.6 | 56.8 | 0.739 | 0.734 | 36.4 | 30.5 | 0.912 | 0.888 |
| $AvgFlow_{\text{DiT-D}}$ (52M) | 1 | **76.8** | **82.8** | **0.548** | **0.541** | **61.0** | **64.0** | **0.720** | **0.675** |

## 4.4. When is Reflow Really Necessary?

From the benchmark results on GEOM-Drugs and GEOM-QM9, we understand that our models after reflow/distillation can achieve better performance than cheminformatics methods on all metrics. However, it is evident that the models' performance drops after reflow especially for the precision metrics. Flow-matching models generally have high generation quality with fewer steps compared to denoising diffusion models (Lipman et al., 2023), thanks to the ODE sampling process. In this section, we are trying to answer the question: when is reflow really necessary to generate high-quality molecular conformers? We address this question with a case study of the NequIP model.

Fig. 3 shows the the performance of our NequIP model using Euler solver with number of steps $N_{\text{step}} \in \{1, 2, 3, 5, 10, 20, 50, 100\}$. The performance of the models is evaluated with the same four metrics on a subset of the GEOM-Drugs test set containing 300 molecules. Overall, $AvgFlow_{\text{NequIP}}$ performs better when $N_{\text{step}} \geq 10$ than $AvgFlow_{\text{NequIP-R}}$. When $N_{\text{step}} < 10$, the performance of $AvgFlow_{\text{NequIP}}$ starts to collapse and eventually reaches 0% coverage for both recall and precision when $N_{\text{step}} = 1$.

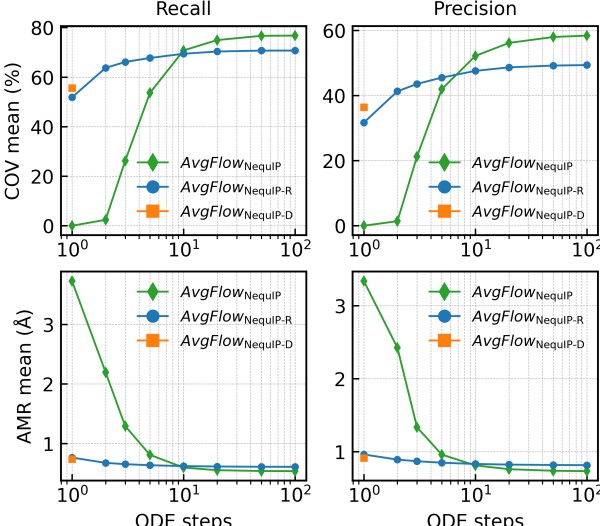

*Figure 3.* **Effect of the number of ODE steps on model's performance.** Comparison between the performance of $AvgFlow_{\text{NequIP}}$ before and after reflow with different number of ODE steps.

The performance gap becomes significant for all metrics when $N_{\text{step}} < 5$. $AvgFlow_{\text{NequIP-R}}$, on the other hand, has

minimal loss in performance until $N_{\text{step}} = 2$ due to the straightened flow trajectory. However, the one-step generation quality of the model still suffers even after reflow. Distillation can effectively reduce the RMSD of one-step generated conformers and improve both the COV-R and COV-P. In summary, reflow is critical when generating molecular conformers with very few ODE steps ($N_{\text{step}} < 5$).

### 4.5. Sampling Time

To demonstrate the sampling efficiency of our model, we first compare the single function call wall-clock time of our model with several strong baselines. Fig. 4 shows that both $AvgFlow_{\text{NequIP}}$ and $AvgFlow_{\text{DiT}}$ are significantly faster than other baselines models in terms of function call time. $AvgFlow_{\text{DiT-L}}$ with 64M parameters is comparable to MCF-S and slightly faster than ET-Flow. The major speed-up of the our model is due to the JAX implementation and a smaller number of parameters (NequIP). Table 6 show that the average sampling time of $AvgFlow_{\text{NequIP-R}}$ (2 steps) for each conformer in the GEOM-Drugs test set is 2.68 microseconds, which is 21 to 50× faster than different variants of MCF sampled with DDIM for 3 steps. It is also 48× faster than torsional diffusion sampled with 5 steps. Meanwhile, $AvgFlow_{\text{NequIP-R}}$ (2 steps) outperforms torsional diffusion and MCF-S (3 steps) by a large margin with only a fraction of the sampling time. Moreover, $AvgFlow_{\text{DiT-D}}$ is faster than the MCF variants (3 steps) with better generation performance in all metrics. $AvgFlow_{\text{DiT-D}}$ also has better recall metrics performance than ET-Flow (2 steps) with ∼3× faster sampling. With reflow/distillation finetuning that ensures high-quality generation with only 2 or even 1 ODE step, our models achieve extraordinary sampling efficiency.

rectness" of generated samples is invariant to rotation (e.g. protein structure generation). The effectiveness of *Averaged Flow* objective is *architecture-agnostic*, meaning that it can be applied to both equivariant and non-equivariant architectures, as demonstrated through experiments. We have also applied the reflow and distillation techniques to straighten the flow trajectory and enable few-step or even one-step molecular conformer generation. Our models reach state-of-the-art performance on the GEOM-QM9 dataset. Our $AvgFlow_{\text{DiT}}$ model matches the performance of other strong baselines and our $AvgFlow_{\text{DiT-L}}$ model achieves state-of-the-art precision metrics performance on the GEOM-Drugs dataset. By analyzing the effect of number of ODE steps on model generation quality, we find that reflow and distillation are necessary when few-step ($N_{\text{step}} < 5$) conformer generation is desired. Most importantly, our $AvgFlow_{\text{DiT-D}}$ model significantly outperforms both diffusion/flow-based and cheminformatic baselines in one-step conformer generation. Combining efficient implementation and one-step generation capability, our method sheds light on the path toward highly efficient yet accurate conformer generation using flow-based generative models. Overall, our method bridges the gap between diffusion/flow-based models and practical molecular conformer generation application by pushing the boundary of quality-speed trade-off.

## Impact Statement

The method proposed in this work improves the training and sampling efficiency of flow-based molecular conformer generation models, advancing the application of deep generative model in the field of computational chemistry and drug discovery. The *Averaged Flow* objective bears broader impact of improving flow-based models for other similar applications such as protein design. Direct consequences of our work include more accurate molecular property prediction and faster compound virtual screening, which lead to societal impacts like accelerated drug discovery and molecular design for environmental purpose.

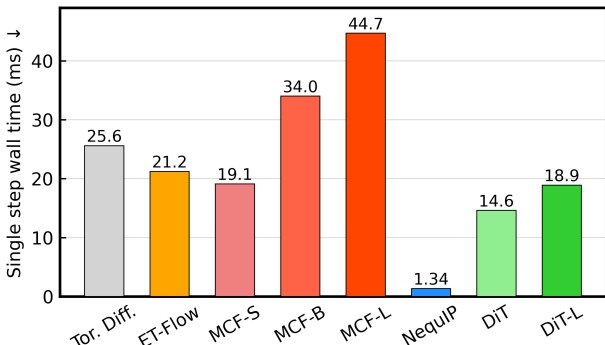

*Figure 4.* **Single function call wall-clock time comparison.**

## 5. Conclusion

We have presented SO(3)-*Averaged Flow* as a new objective to accelerate the training of flow-matching models for molecular conformer generation. *Averaged Flow* leads to faster convergence and better performance compared with conditional OT and Kabsch alignment. The *Averaged Flow* objective can also be extended to other applications when the "cor-

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

# Appendix

## A. Details of Model Architectures

### A.1. Modified NequIP Architecture

The equivariant model used in this work is a modified variant (Fig. 5) of the NequIP model (Batzner et al., 2022). The model takes 5 inputs including the atomic features $\mathbf{Z}$, coordinate of atoms $x_t$, relative distance vector between atoms $\vec{r}$, edge (bond) features $\mathbf{E}$, and the flow-matching timestep $t$. The output model is a vector field corresponding to the probability flow at $t$. Compared to the original NequIP model, our variant has residue connection and equivariant layer normalization (Liao et al., 2023) after each interaction block, which we found to be highly effective in stabilizing the training of model with more than 4 layers. Bond information in the 2D molecular graph is critical inductive bias for the molecular conformer generation task. To add bond information into the model, we featurize the edges in the molecular graph and concatenate the edge features $\mathbf{E}$ with the radial basis embedding of relative distance vector $\vec{r}$. The concatenated message is then fed into the rotationally invariant radial function implemented as an multi-layer perceptron. To keep long-range information in the graph convolution during intermediate time-step $t$, we remove the envelop function from the radial basis and keep only the radial Bessel function.

For both the GEOM-Drugs and GEOM-QM9 dataset, we train a model with 6 interaction blocks. The multiplicity is set to 96 and maximum order of irreps $l$ is 2. The radial function MLP has 2 layers and hidden dimension of 256. Molecular graph are fully-connected with non-bond as an specified bond type. The relative distance vectors are scaled down by a soft cutoff distance of 10Å and 20Å for QM9 and Drugs dataset, respectively. we used 12 Bessel radial basis functions in the model. The model is implemented using `e3nn-jax` (Geiger & Smidt, 2022; Geiger et al., 2022).

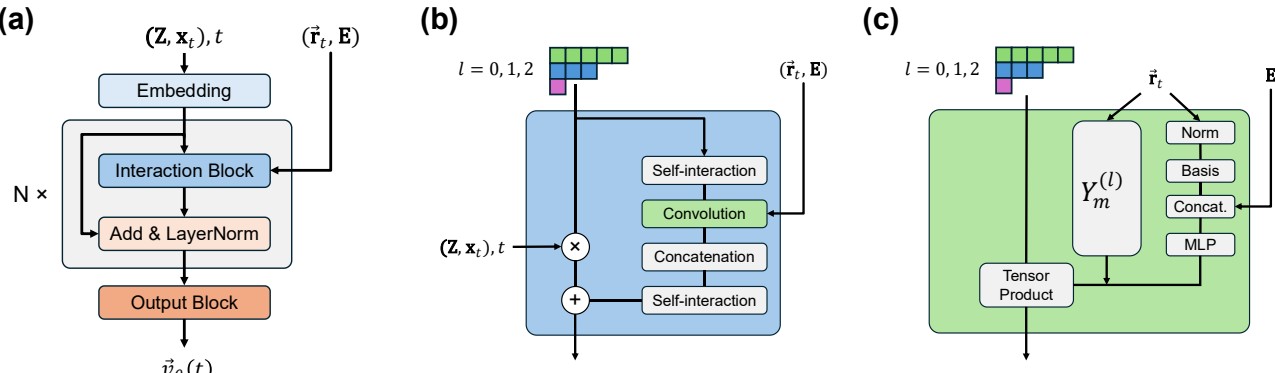

*Figure 5.* **Modified NequIP model architecture** *(a)* Overview of the modified NequIP architecture for the flow vector field prediction. *(b)* Details of the interaction block, where atomic features are mixed and refined with relative distance vectors $\vec{r}$ and edge features $\mathbf{E}$. *(c)* In the convolution block, a learnable radial function MLP incorporate basis embedding of $\vec{r}$ and edge features $\mathbf{E}$. Tensor product is used to combine the output of the MLP and the spherical harmonics $Y_m^{(l)}$ projection of $\vec{r}$.

### A.2. DiT Architecture

The diffusion transformer (Peebles & Xie, 2023) is a powerful yet scalable architecture for generative modeling. Inspired by recent successes of such architecture in protein structure prediction (Abramson et al., 2024) and generation(Geffner et al., 2025), we implemented the diffusion with pairwise attention bias (DiT) model and modified it specifically for molecular conformer generation task. Fig. 6 shows a schematic of the model and the details of the main trunk of the architecture. Besides the adaptive bias and scaling used in standard diffusion transformer, we also added: *(i)* auxiliary register tokens (Xiao et al., 2023; Darcet et al., 2023) and *(ii)* QK normalization (Dehghani et al., 2023) for the training stability. The inputs to the model are the same input features to the NequIP model. The atomic features $\mathbf{Z}$ are concatenated with the coordinate of atoms $x_t$ and project to atomic representation of each atom. A sinusoidal embedding of the timestep $t$ is used as the condition embedding. The pairwise distance $\|\vec{r}\|$ between atoms and the bond features $\mathbf{E}$, which is embedded from 5 bond

types Sec.B.2, are used to construct a pairwise representation, which is then injected as bias into the attention. Gating mechanism applied over the output of attention block to control the update of the atomic representation. In this work, we trained two variants of the DiT model: DiT (52M params.) and DiT-L (64M params.). The hyperparameters of the two variants are tabulated in Table. 3.

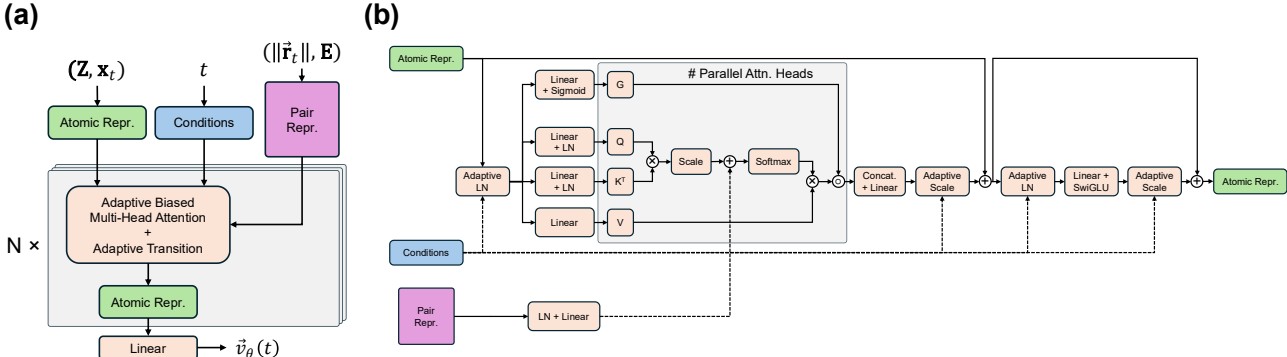

Figure 6. **DiT model architecture.** *(a)* The overview of DiT. *(b)* The details of the adaptive multi-head attention with pairwise bias and adaptive transition block.

Table 3. **Hyperparameters for the architecture of the DiT models.**

| Model | DiT | DiT-L |
|---|---|---|
| **Architecture Component** | | |
| initialization | random | random |
| atomic repr. dim | 512 | 576 |
| # register tokens | 10 | 10 |
| cond. dim | 512 | 196 |
| $t$ sinusoidal enc dim | 512 | 196 |
| pair repr. dim | 128 | 128 |
| # attention heads | 8 | 12 |
| # transformer layers | 8 | 10 |
| # trainable parameters | 52M | 64M |

# B. Experiments Details

## B.1. Datasets

The dataset we train and benchmark our model on are GEOM-Drugs and GEOM-QM9(Axelrod & Gomez-Bombarelli, 2022). We follow the exact splitting defined and used in previous works (Ganea et al., 2021; Jing et al., 2022; Wang et al., 2024). The train/val/test set of GEOM-Drugs contains 243473/30433/1000 molecules, respectively. The train/val/test set of GEOM-QM9 contains 106586/13323/1000 molecules, respectively.

## B.2. Molecular Graph Featurization

We followed the atomic featurization from GeoMol (Ganea et al., 2021). Details of the atomic featurization are included in Table. 4. Graph Laplacian positional encoding vector (Dwivedi et al., 2023) with size of 32 is concatenated with the atomic features for each atom in molecular graph to form the final atomic feature vector $\mathbf{Z}$. The edge features is the one-hot encoding of the bond types: {No Bond, Single Bond, Double Bond, Triple Bond, Aromatic Bond}.

*Table 4.* Atomic features as input to the model

| Name | Description | Range |
|------|-------------|-------|
| atom_type | Atom type | One-hot encoding of the atom type |
| degree | Number of bonded neighbors | $\{x : 0 \leq x \leq 6, x \in \mathbb{Z}\}$ |
| charge | Formal charge of atom | $\{x : -1 \leq x \leq 1, x \in \mathbb{Z}\}$ |
| valence | Implicit valence of atom | $\{x : 0 \leq x \leq 6, x \in \mathbb{Z}\}$ |
| hybridization | Hybridization type | $\{\text{sp}, \text{sp}^2, \text{sp}^3, \text{sp}^3\text{d}, \text{sp}^3\text{d}^2, \text{other}\}$ |
| chirality | Chirality Tag | {unspecified, tetrahedral CW, tetrahedral CCW, other} |
| num_H | Total number of hydrogens | $\{x : 0 \leq x \leq 8, x \in \mathbb{Z}\}$ |
| aromatic | Whether on aromatic ring | {True, False} |
| num_rings | Number of rings the atom on | $\{x : 0 \leq x \leq 3, x \in \mathbb{Z}\}$ |
| ring_size_3-8 | Whether on ring size of 3-8 | {True, False} |

## B.3. Training and Sampling Details

### B.3.1. NEQUIP TRAINING AND SAMPLING

The NequIP model is trained with the *Averaged Flow* for 990 epochs on the GEOM-Drugs dataset and 1500 epochs on the GEOM-QM9 dataset using 2 NVIDIA A5880 GPUs. We used dynamic graph batching to maixmize the utilization of GPU memory and reduce JAX compilation time. The effective average batch size is 208 and 416 for Drugs and QM9 dataset, respectively. We used Adam optimizer with learning rate of $1\mathrm{e}{-2}$, which decays to $5\mathrm{e}{-3}$ after 600 epochs and to $1\mathrm{e}{-3}$ after 850 epochs. We selected the top-30 conformers for model training.

To sample coupled $(x'_0, x'_1)$ for reflow and distillation, we generate 32 noise-sample pairs for each molecule in the Drugs and 64 for each molecule in the QM9 dataset. The reflow and distillation are done using 4 NVIDIA A100 GPUs and doubling the effective batch size of each dataset. During the reflow stage, the model is finetuned for 870 epochs on Drugs and 1530 epochs on QM9. We used Adam optimizer with learning rate of $5\mathrm{e}{-3}$, which decays to $2.5\mathrm{e}{-3}$ after 450 epochs for Drugs (500 epochs for QM9), and to $5\mathrm{e}{-4}$ after 650 epochs for Drugs (900 epochs for QM9). During the distillation stage, the model is finetuned for 450 epochs on Drugs and 1200 epochs on QM9. We used Adam optimizer with learning rate of $2\mathrm{e}{-3}$, which decays to $1\mathrm{e}{-3}$ after 300 epochs for Drugs (500 epochs for QM9), and to $2\mathrm{e}{-4}$ after 450 epochs for Drugs (900 epochs for QM9). We used exponential moving average (EMA) with a decay of 0.999 for all *Averaged Flow*, reflow, and distillation training.

To generate the benchmark results of $AvgFlow_{\text{NequIP}}$ (Table. 1 and Table. 2), we use the Tsitouras' 5/4 solver (Tsitouras, 2011) implemented in the diffrax package with adaptive stepping. The relative tolerance and absolute tolerance are set to $1\mathrm{e}{-5}$ and $1\mathrm{e}{-6}$ when sampling for GEOM-Drugs, respectively. The relative tolerance and absolute tolerance are both set to $1\mathrm{e}{-5}$ when sampling for GEOM-QM9. Euler solver is always used for $AvgFlow_{\text{NequIP-R}}$ and $AvgFlow_{\text{NequIP-D}}$. When comparing the effect of ODE steps to models, Euler solver is used.

### B.3.2. DiT TRAINING AND SAMPLING

We trained 2 variants of the DiT model, DiT (52M) and DiT-L (64M), using *Averaged Flow*. We used the cosine learning rate decay with warm-up steps for learning adjustment and AdamW optimizer (Loshchilov & Hutter, 2017). The details of hyperparameters in training are tabulated in Table. 5. Due to computational cost, we only did reflow and distillation finetuning of the 52M DiT model on the GEOM-Drugs dataset. 32 noise-sample pairs were generated for each molecule in the Drugs using 100 Euler steps to construct the reflow/distillation dataset. We used EMA with a decay of 0.999 for all *Averaged Flow*, reflow, and distillation training. We used 100 Euler steps to generate conformers for all benchmarks of $AvgFlow_{\text{DiT}}$.

Table 5. **Hyperparameters for DiT models training.**

| Model | DiT (GEOM-Drugs) | DiT-L (GEOM-Drugs) | DiT (GEOM-QM9) |
|---|---|---|---|
| **Training Details** | | | |
| # train epochs | 760 | 900 | 900 |
| batch size per GPU | 64 | 64 | 256 |
| # GPUs | 8 | 8 | 2 |
| GPU name | NVIDIA A100 | NVIDIA A100 | NVIDIA A5880 |
| optimizer | AdamW | AdamW | AdamW |
| init learning rate | 1e−6 | 1e−6 | 1e−6 |
| peak learning rate | 2e−4 | 2e−4 | 2e−4 |
| warm-up steps | 10k | 10k | 10k |
| cosine lr decay steps | 1M | 1M | 1M |
| ending learning rate | 1e−6 | 1e−6 | 1e−6 |
| reflow epochs | 260 | - | - |
| reflow peak learning rate | 1e−4 | - | - |
| distill epochs | 130 | - | - |
| distill peak learning rate | 5e−5 | - | - |

### B.4. Chirality Correction

A simple post-hoc chirality correction step is used during sampling. When the conformer of a molecule that contains at least 1 chiral center atom is generated, we use RDKit (Landrum, 2016) to perform the chirality correction withe following steps: *(i)* Given a `Mol` object with generated conformer, re-assign chiral tag to atoms. *(ii)* Obtain the the canonical SMILES (Weininger, 1988) of the `Mol` object and compare it to the original canonical SMILES. *(iii)* If there is a mismatch of the two canonical SMILES strings, we mirror the generated conformer. The post-hoc chirality correct is a simple, fast, and end-to-end solution to the rare chirality mismatch during generation.

### B.5. Distribution of $t$ during Reflow

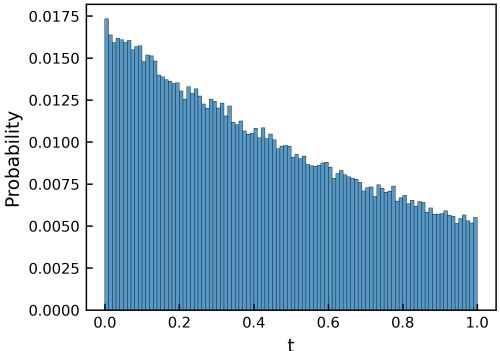

Figure 7. **The distribution of $t$ during reflow**

The distribution of $t$ during reflow is sampled from $p(t) \propto \text{Exp}(\lambda t)$, where $\lambda = -1.2$. The distribution is visualized in Fig. 7.

## B.6. Evaluation Metrics

We report the average minimum RMSD (AMR) between ground truth and generated structures, and Coverage for Recall and Precision. Coverage is defined as the percentage of conformers with a minimum error under a specified AMR threshold. Recall matches each ground truth structure to its closest generated structure, and Precision measures the overall spatial accuracy of the each generated structure. Following Ganea et al. (2021); Jing et al. (2022), we generate two times the number of ground truth structures for each molecule. More formally, for $K = 2L$, let $\{C_l^*\}_{l \in [1,L]}$ and $\{C_k\}_{k \in [1,K]}$ respectively be the sets of ground truth and generated structures:

$$
\begin{aligned}
\text{COV-Precision} &:= \frac{1}{K} \left| \{k \in [1..K] : \min_{l \in [1..L]} \text{RMSD}(C_k, C_l^*) < \delta\} \right|, \\
\text{AMR-Precision} &:= \frac{1}{K} \sum_{k \in [1..K]} \min_{l \in [1..L]} \text{RMSD}(C_k, C_l^*),
\end{aligned}
\tag{13}
$$

where $\delta$ is the coverage threshold. $\delta$ is set to $0.75$Å for the Drugs and $0.5$Å for the QM9 dataset. The recall metrics are obtained by swapping ground truth ($K$) and generated conformers ($L$) in the above equations.

## B.7. ODE Trajectory Visualization

In this section, we visualize the ODE trajectory of a few selected test set molecules. For each molecule, the 100-steps ODE trajectory of the $AvgFlow_{\text{DiT}}$ model before reflow is shown as orange curves, and the 1-step trajectory of the $AvgFlow_{\text{DiT-D}}$ model after reflow+distillation is shown as green curves.

## B.8. More Comparison of Few-step Generation Performance

Here we provide compare the inference time of models on few-step generation while maintaining *reasonable* generation quality.

*Table 6.* **Sampling time and performance comparison between models for few-step sampling.** In this table, we record the sampling time of baselines and our models with the minimum number of steps to generate *reasonably* good conformers. Mean are reported for all metrics. Bold results are the best.

| Method | Step | Time (ms) ↓ | COV-R (%) ↑ | AMR-R (Å) ↓ | COV-P (%) ↑ | AMR-P (Å) ↓ |
|---|---|---|---|---|---|---|
| Tor. Diff. | 5 | 128 | 58.4 | 0.691 | 36.4 | 0.973 |
| ET-Flow | 5 | 106 | **77.8** | **0.476** | **74.0** | **0.550** |
| ET-Flow | 2 | 42.8 | **73.2** | **0.577** | **63.8** | **0.681** |
| MCF-S | 3 | 57.3 | 56.9 | 0.725 | 30.8 | 1.014 |
| MCF-B | 3 | 102 | 66.5 | 0.665 | 39.9 | 0.951 |
| MCF-L | 3 | 134 | 71.6 | 0.636 | 45.3 | 0.686 |
| $AvgFlow_{\text{NequIP-R}}$ | 2 | **2.68** | 64.2 | 0.663 | 43.1 | 0.871 |
| $AvgFlow_{\text{DiT-D}}$ | 1 | **14.6** | 76.8 | 0.548 | 61.0 | 0.720 |

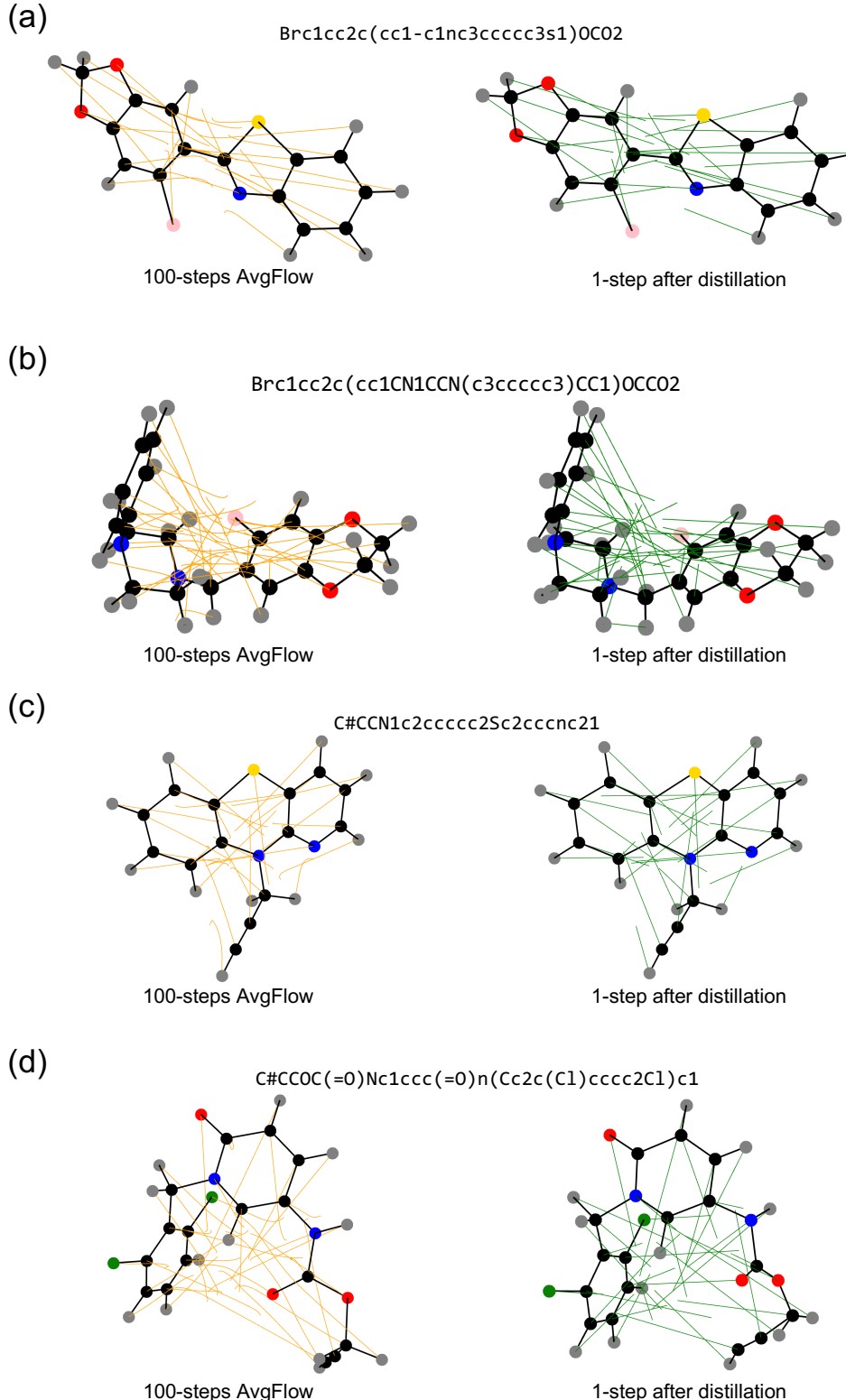

*Figure 8.* **Comparison between ODE trajectories.** Visualization of selected generated conformers (SMILES attached) and the ODE trajectories. Orange trajectories are from *AvgFlow*$_{\text{DiT}}$ before reflow, and green trajectories are from *AvgFlow*$_{\text{DiT-D}}$ after reflow+distillation. Carbon atoms are colored as black, hydrogen as gray, oxygen as red, nitrogen as blue, sulfur as yellow, clorine as green, and boron as pink.

# C. Averaged Flow Details

## C.1. Python Implementation

In this section, we provide the Python implementation of the *Averaged Flow* training objective as supplmentary information to Sec.3.1.

*Listing 1.* Averaged Flow

```python
def avg_harmonic_flow(
    t: jax.Array,  # []
    x: jax.Array,  # [num_nodes, 3]
    x1: jax.Array,  # [num_conformers, num_nodes, 3]
    edges: jax.Array,  # [2, num_edges]
    weights: jax.Array | None = None,  # [num_conformers]
    sigma0: jax.Array = 1.0,
    sigma1: jax.Array = 0.0,
) -> jax.Array:
    degree = jnp.bincount(edges[0], length=x.shape[0])

    def metric(x, y):
        # x and y have shape [num_nodes]
        sigma_t = (1 - t) * sigma0 + t * sigma1
        laplacian = (
            jnp.sum(degree * x * y)
            - jnp.sum(x[edges[0]] * y[edges[1]])
            - jnp.sum(x[edges[1]] * y[edges[0]])
        )
        return laplacian / sigma_t**2

    avg_x1 = avg_target(x, x1, t, metric, weights)

    return (avg_x1 - x) / (1 - t)

def avg_flow(
    t: jax.Array,  # []
    x: jax.Array,  # [num_nodes, 3]
    x1: jax.Array,  # [num_conformers, num_nodes, 3]
    weights: jax.Array | None = None,  # [num_conformers]
    sigma0: jax.Array = 1.0,
    sigma1: jax.Array = 0.0,
) -> jax.Array:
    def metric(x, y):
        # x and y have shape [num_nodes]
        sigma_t = (1 - t) * sigma0 + t * sigma1
        return jnp.dot(x, y) / sigma_t**2

    avg_x1 = avg_target(x, x1, t, metric, weights)

    return (avg_x1 - x) / (1 - t)

def avg_target(
    x: jax.Array,  # [n, 3]
    targets: jax.Array,  # [num_targets, n, 3]
    t: jax.Array,  # []
    metric: Callable[[jax.Array, jax.Array], jax.Array],
    weights: jax.Array | None = None,  # [num_targets]
) -> jax.Array:
    num_targets, n, _ = targets.shape
    assert x.shape == (n, 3)
    assert targets.shape == (num_targets, n, 3)
    assert t.shape == ()

    def outer(u, v):  # [n, 3] x [n, 3] -> [3, 3]
        return jax.vmap(jax.vmap(metric, (None, -1)), (-1, None))(u, v)

    def inner(u, v):  # [n, 3] x [n, 3] -> []
        return jnp.sum(jax.vmap(metric, (-1, -1))(u, v))

    def logZ(alpha):  # [n, 3] -> []
        def f(target):  # [n, 3] -> []
            return (
                logcF(t * outer(target, x) + target.T @ alpha)
                - (inner(x, x) + t**2 * inner(target, target)) / 2
            )

        return logsumexp(jax.vmap(f)(targets), weights)

    return jax.grad(logZ)(jnp.zeros_like(x))

def logsumexp(a: jax.Array, weights: jax.Array | None = None) -> jax.Array:
    assert a.ndim == 1
    assert weights is None or weights.shape == a.shape
    where = (weights > 0) if weights is not None else None

    amax = jnp.max(a, where=where, initial=-jnp.inf)
```

```python
        amax = jax.lax.stop_gradient(
            jax.lax.select(jnp.isfinite(amax), amax, jax.lax.full_like(amax, 0))
        )
        if where is not None:
            a = jnp.where(where, a, amax)
        exp_a = jax.lax.exp(jax.lax.sub(a, amax))
        if weights is not None:
            exp_a = exp_a * weights
        sumexp = exp_a.sum(where=where)
        return jax.lax.add(jax.lax.log(sumexp), amax)

# All the code below is adapted from a PyTorch code from David Mohlin, Gerald Bianchi and Josephine Sullivan

def logcF(F: jax.Array) -> jax.Array:
    # \log \int_{SO(3)} \exp(\text{tr}(F^T R)) dR
    assert F.shape == (3, 3)
    return logcf(*signed_svdvals(F))

def signed_svdvals(F: jax.Array) -> jax.Array:
    u, s, vh = jnp.linalg.svd(F, full_matrices=False)
    u, vh = jax.lax.stop_gradient((u, vh))
    sign = jnp.sign(jnp.linalg.det(u @ vh))
    return s.at[-1].mul(sign)

@jax.custom_vjp
def logcf(s1: jax.Array, s2: jax.Array, s3: jax.Array) -> jax.Array:
    # assume s1 >= s2 >= s3
    s1, s2, s3 = jnp.asarray(s1), jnp.asarray(s2), jnp.asarray(s3)
    return s1 + s2 + s3 + jnp.log(factor(False, s1, s2, s3))

def _logcf_fwd(
    s1: jax.Array, s2: jax.Array, s3: jax.Array
) -> tuple[jax.Array, tuple[jax.Array, jax.Array]]:
    # s1 >= s2 >= s3
    f = factor(False, s1, s2, s3)
    return s1 + s2 + s3 + jnp.log(f), (s1, s2, s3, f)

def _logcf_bwd(res: tuple[jax.Array, ...], grad: jax.Array) -> tuple[jax.Array]:
    s1, s2, s3, f = res
    # s1 >= s2 >= s3
    assert s1.shape == ()
    assert f.shape == ()
    assert grad.shape == ()
    g1 = grad * factor(True, s1, s2, s3) / f
    g2 = grad * factor(True, s2, s1, s3) / f
    g3 = grad * factor(True, s3, s1, s2) / f
    return g1, g2, g3

logcf.defvjp(_logcf_fwd, _logcf_bwd)

def factor(add_x: bool, s1: jax.Array, s2: jax.Array, s3: jax.Array) -> jax.Array:
    def f(x):
        i0 = (1.0 - 2 * x) if add_x else 1.0
        i1 = bessel0((s2 - s3) * x)
        i2 = bessel0((s2 + s3) * (1 - x))
        return i0 * i1 * i2

    tiny = jnp.finfo(s1.dtype).tiny
    a = 2 * (s3 + s1)

    # a non zero:
    a_ = jnp.maximum(a, 0.5)
    y = jnp.linspace(tiny + jnp.exp(-a_), 1.0, 512)
    r1 = jnp.trapezoid(jax.vmap(f)(-jnp.log(y) / a_), y) / a_

    # a (close to) zero:
    x = jnp.linspace(0.0, 1.0, 512, dtype=s1.dtype)
    r2 = jnp.trapezoid(jax.vmap(f)(x) * jnp.exp(-a * x), x)

    return jnp.where(a > 1.0, r1, r2)

def bessel0(x: jax.Array) -> jax.Array:
    p = [1.0, 3.5156229, 3.0899424, 1.2067492, 0.2659732, 0.360768e-1, 0.45813e-2]
    bessel0_a = jnp.array(p[::-1], dtype=x.dtype)

    p = [0.39894228, 0.1328592e-1, 0.225319e-2, -0.157565e-2, 0.916281e-2]
    p += [-0.2057706e-1, 0.2635537e-1, -0.1647633e-1, 0.392377e-2]
    bessel0_b = jnp.array(p[::-1], dtype=x.dtype)

    abs_x = jnp.abs(x)
    x_lim = 3.75

    def w(x, y):
        return jnp.where(abs_x <= x_lim, x, y)
```

```
    abs_x_ = w(x_lim, abs_x)

    return w(
        jnp.polyval(bessel0_a, w(abs_x / x_lim, 1.0) ** 2) * jnp.exp(-abs_x),
        jnp.polyval(bessel0_b, w(1.0, x_lim / abs_x_)) / jnp.sqrt(abs_x_),
    )
```

## C.2. Speed Benchmark

We benchmarked the time used by our Python implementation to solve the *Averaged Flow* objective for batched graphs. Each graph is set to have 50 nodes (the average number of atoms in GEOM-Drugs molecules is 44). The benchmark is done on a single NVIDIA A5880 GPU.

*Table 7.* Computation time of *Averaged Flow* on batched graphs (50 nodes per graph). Unit is in ms. $N_{\text{batch}}$ is the number of graphs in a batch and $N_{\text{conformer}}$ is number of conformers used in *Averaged Flow* solving.

| $N_{\text{batch}}$ \ $N_{\text{conformer}}$ | 1 | 10 | 100 | 1000 |
|---|---|---|---|---|
| 1 | 0.6 | 0.5 | 0.5 | 0.6 |
| 10 | 0.5 | 0.5 | 0.6 | 1.0 |
| 100 | 0.5 | 0.6 | 1.1 | 7.6 |
| 1000 | 0.5 | 0.9 | 7.5 | 73.5 |

