# OpenReview forum: "Efficient Molecular Conformer Generation with SO(3)-Averaged Flow Matching and Reflow"
_ICML.cc/2025/Conference — ICML 2025 poster_

### Official Review · Reviewer_iSgP · 2025-02-18

**Overall Recommendation:** 3

**Summary:**

This paper focuses on improving the training and inference efficiency of 3D molecular conformer generation while matching the performance of strong baselines.
To improve training efficiency, it introduces a new training objective, called SO(3)-Averaged Flow, which can avoid the need for rotational alignment between prior and data distribution by training the model to learn the average probability path over all rotations of the data.
Then, it further introduces the reflow and distillation technique for fast inference, which can achieve high-quality molecular conformer generation with few-step or even one-step ODE solving.

**Claims And Evidence:**

One significant limitation of this paper is that many claims are experimental and empirical while lacking clear and convincing theoretical evidence.
For example, section 3.1 lacks further details on the derivation of the relevant formulas.
It's difficult for me to understand all these formulas without additional information and references, and I want to know whether they are supported by theoretical evidence.

**Essential References Not Discussed:**

None.

**Experimental Designs Or Analyses:**

I think the experimental designs and analyses are sound for this problem.

**Methods And Evaluation Criteria:**

As a novel concept, I think the proposed SO(3)-Averaged Flow makes sense for this problem.

**Other Comments Or Suggestions:**

None.

**Other Strengths And Weaknesses:**

Weaknesses:

1.  As shown in Table 1 and Table 2, the performance of the proposed method is significantly weaker than strong baselines. In this case, the meaning of efficiency improvement is limited.

2. The sampling efficiency comparison in Table 3 is unfair.
    * Why not implement these baselines with the same steps? Reviewers need to know whether these baselines can still perform better with the same steps.
    * The fair comparison should be entirely based on the methods themselves, and the influence of their specific implementation (e.g., JAX implementation) needs to be removed.

**Questions For Authors:**

1. Why not increase the model size to match those strong baselines? After all, it will be more convincing if the proposed method can achieve comparable or even better performance.

2. Can you provide the relevant experimental support for the model architecture independence of the reflow and distillation algorithm?

**Relation To Broader Scientific Literature:**

Since the key contributions of the paper are built upon flow-matching, which aims to improve the training and sampling efficiency of flow-based models, it's possible to bear an impact on other similar applications base on flow-matching, such as protein design.

**Theoretical Claims:**

The theoretical claims of this paper are mainly concentrated in section 3.1. However, it's difficult for me to check their correctness without further details.

---

> ### Author Rebuttal · Authors · 2025-04-01
>
> *We would like to thank the reviewer for reviewing and acknowledging the novelty of the SO(3)-Averaged Flow. Please see the response below:*
>
> **Theoretical Claims**
>
> The major motivation behind the development of *AvgFlow* is to eliminate the need for data augmentation through rotation by training the model to learn the flow from interpolant $x_t$ to ground truth $x_1$ averaged over the SO(3) group. Sec 3.1 is the major contribution, as it mathematically derives a closed-form solution for the SO(3)-Averaged Flow so that training can be efficient without much of computational overhead. We have also attached the Python implementation of solving the *AvgFlow* objective in Sec A.3 to accompany the mathematical derivation. We would be happy to take further questions from the reviewer about specific steps in Sec. 3.1.
>
> **Weakness**
>
> 1. We want to emphasize that the major motivation of this paper is to improve the efficiency of diffusion/flow-based models to the level of cheminformatics tools for conformer generation through algorithmic innovations. Given the need of ultra-large scale (~$10^8$-$10^9$ compounds) virtual screening, diffusion/flow conformer generation model would become practical with improved sampling efficiency. Bearing the motivation in mind, we chose to implement a compact 4.7M-parameter equivariant GNN model that achieves significant sampling speedup while maintaining good generation quality. Comparing to larger transformer-based baselines, the model's speedup in few-step sampling compensates for reduced generation quality: the 2-step $\mathrm{AvgFlow_Reflow}$ model outperforms 3-step MCF-B in precision metrics. It also achieves ~58% of the 5-step ETFlow with ~40x speedup in sampling. We believe that the current 4.7M model has fulfilled this motivation.
>
>    That being said, we agree with the reviewer that achieving the SOTA in generation quality is valuable. We have implemented a more scalable diffusion transformer (DiT) with pairwise biased attention, similar to AlphaFold3 [1] and Prote&iacute;na [2], for conformer generation. Due to the limited time, we trained a 52M-parameter DiT model with *AvgFlow* objective for only ~124k steps and benchmarked it on the Drugs test set (see results [here](https://anonymous.4open.science/r/confgen_icml25_rebuttal-8406/benchmark.png)). We have also compared the performance of the DiT model trained with *AvgFlow*, Kabsch, and Conditional OT (same experiment as Sec 4.1 of manuscript, see results [here](https://anonymous.4open.science/r/confgen_icml25_rebuttal-8406/dit_obj_comp.png)). The benchmark results of DiT demonstrates that:
>
>     - *AvgFlow* can be a better training objective than Conditional OT and Kabsch even with **non-equivariant model architecture**.
>
>     - A transformer-based architecture with more parameters trained with *AvgFlow* can achieve performance on-par with other SOTA models, even with a relatively limited number of training steps.
>
>    The performance of the DiT model is expected to improve further as it is trained for more steps. It can also be scaled up to similar size to MCF-L for better performance if resources allow. We are currently working on reflow fine-tuning the DiT model and we will share results once completed.
>
> 1. Clarification about Table 3:
>    - We agree with the reviewer that baselines should be compared with same number of sampling steps when possible. We took the benchmark values from corresponding papers because their checkpoints were not released before this manuscript was submitted. We have now benchmarked the 2-step and 1-step generation results of ETFlow and MCF-B/L (see updated Table 3 here). Our reflow model outperforms MCF for 2-step generation across all metrics. More importantly, our distill model outperforms both ETFlow and MCF by a large margin for 1-step generation, especially in the recall metrics. Despite the high 2-step generation quality of ETFlow, our reflow and distill models are still 16x and 32x faster in sampling, respectively.
>
>    - We want to respectfully disagree with the reviewer's suggestion that the effect of implementation should be excluded when comparing sampling speed. We would argue that faster implementation should be counted as a technical contribution as it validates a potential direction for optimization and acceleration.
>
> **Answers to questions**
> 1. Please see above answers regarding experiments with larger model.
>
> 1. Yes, we are working on reflow/distill fine-tuning the DiT model (non-equivariant transformer architecture) mentioned above. These experiments can be time-consuming as they require generating ($X_0'$, $X_1'$) couples. We will report results once completed.
>
> References:
>
> [1] Abramson et al. "Accurate structure prediction of biomolecular interactions with AlphaFold 3." *Nature* (2024)
>
> [2] Geffner et al. "Proteina: Scaling Flow-based Protein Structure Generative Models." *ICLR* (2025).

---

> > ### Comment · Reviewer_iSgP · 2025-04-08
> >
> > Apologies for the delayed response.
> >
> > However, I remain concerned about the performance of AvgFlow. As shown in Table 6, 50-step ET-Flow (8.3M) can still achieve on-par performance than 100-step AvgFlow (52M). What's the advantage of AvgFlow in this context? Can 100-step AvgFlow (52M) still maintain better efficiency compared to 50-step ET-Flow (8.3M)?

---

> > > ### Author Response · Authors · 2025-04-08
> > >
> > > Thank you very much for the comment. We understand your concern that the performance of AvgFlow$_{\mathrm{DiT}}$ may degrade with fewer sampling steps (100→50). We want to further address this concern by providing additional benchmarks and clarify of the motivation behind our new flow-matching objective *AvgFlow*.
> > >
> > > Firstly, we benchmark the AvgFlow$_{\mathrm{DiT}}$ with only 50 sampling steps, please see the table below:
> > >
> > > **Quality of generated conformer ensembles for GEOM-DRUGS (δ=0.75Å) test set**
> > > *Coverage (COV) and Average Minimum RMSD (AMR) in both Recall and Precision settings. Values are presented in the format of Mean (Median) for each metric.*
> > > | Method| Step| COV-R (%) ↑| AMR-R (Å) ↓| COV-P (%) ↑| AMR-P (Å) ↓|
> > > |-------|-----|------------|------------|------------|------------|
> > > | **No Step Limit**||||||
> > > | ET-Flow-SS (8.3M)|50|79.6 (84.6)|0.439 (0.406)|75.2 (81.7)|0.517 (0.442)|
> > > | AvgFlow$_{\mathrm{DiT}}$ (52M)|100|82.0 (86.7)|0.428 (0.401)|72.9 (78.4)|0.566 (0.506)|
> > > | AvgFlow$_{\mathrm{DiT}}$ (52M)|50|82.0 (86.6)|0.429 (0.401)|72.8 (78.4)|0.567 (0.506)|
> > >
> > > We can observe that the performance difference between sampling 100 steps and 50 steps is minimal for AvgFlow$_{\mathrm{DiT}}$, which is expected because the learned flow trajectory is fairly smooth with curvatures concentrate primarily at $t<0.5$. A visualization can be found in the of Fig.1b of the manuscript. Comparing the 50 steps results of ET-Flow-SS and our model, they are generally on par because our model is slightly better in the Recall metrics but lags marginally in the Precision metrics.
> > >
> > > We also want to clarify the motivation of developing the *SO(3)-Averaged Flow* objective. In a nutshell, the goal of *AvgFlow* is to __enhance the training efficiency__ of flow-matching models for conformer generation: Faster convergence to better generation performance. This is achieved by analytically averaging the flow from an interpolant $x_t$ to all rotations of target $x_1$. To validate the claim, we benchmarked the per-epoch (from 4-100 epochs) generation performance of two model architectures: NequIP (equivariant GNN with 4.7 params) and DiT (non-equivariant transformer with 52M params). We demonstrated that both models trained with *AvgFlow* converge faster to better performance (Fig.2 of the manuscript for NequIP and [rebuttal figure](https://anonymous.4open.science/r/confgen_icml25_rebuttal-8406/dit_obj_comp.png) for DiT), compared with conditional OT flow and Kabsch-alignment flow which are commonly used. This experiment also showcased that the effectiveness of *AvgFlow* as a training objective is **not** restricted to equivariant architectures.
> > >
> > > Additionally, we want to mention that the sampling efficiency improvement is mostly achieved by straightening the flow using reflow/distillation fine-tuning technique. The one-shot generation quality of AvgFlow$_{\mathrm{DiT-Distill}}$ significantly outperformed all other models (see previous reply to [Reviewer anyM's comment](https://openreview.net/forum?id=6uPcJtMgWN&noteId=gSPr4Hu81O)). It even outperformed Tor. Diff., a strong baseline that starts diffusion from RDKit generated valid conformers.
> > >
> > > We appreciate your constructive feedback and  hope our extra benchmark results and comments clarify your concern regarding the manuscript.

---

### Official Review · Reviewer_Adi6 · 2025-03-12

**Overall Recommendation:** 3

**Summary:**

The paper introduces a new method for molecular conformer generation task called Averaged Flow.  Averaged Flow is an SO(3) Flow Matching method that addresses rotational symmetry in 3D molecular structures by integrating overall SO(3) group transformations during training. The authors combined their approach with rectified flow to reduce the number of sampling steps. The method has been evaluated on two common benchmarks in molecular conformer generations: GEOM-QM9 and GEOM-Drugs datasets.

**Claims And Evidence:**

I did not find enough evidence for the claim: "Averaged Flow leads to faster convergence to better performance for molecular conformer generation, and can be extended to other similar tasks."

The method did not outperform the state of the art, and also the paper about conformer generation.

**Essential References Not Discussed:**

No

**Experimental Designs Or Analyses:**

Yes

**Methods And Evaluation Criteria:**

Yes

**Other Comments Or Suggestions:**

No

**Other Strengths And Weaknesses:**

**Strengths**:

- Integrating reflow and distillation methods to improve the sampling time in conformer generation is novel.
- Ablation studies show the effectiveness of the proposed method, with less sampling time compared to the current SOTA.

**Weaknesses**:

Despite the improvement in sampling time, the paper still has several weaknesses.

- Performance gap: The performance of the proposed method is far from previous approaches in most of the evaluation metrics.

- Need for OOD evaluation: The paper lacks evaluation in out-of-distribution settings. For example, both MCF and ET-Flow have been evaluated on larger molecules dataset (GEOM-XL).

**Questions For Authors:**

I did not understand how integrating over the group orbits gives all the possible conformers. Conformers of a molecule consist of different arrangements of atoms in 3D space, not just the result of direct rotation of the whole molecule. Also, if we have a molecule x, for example, and a rotated molecule g.x, should they not have different energy states?

**Relation To Broader Scientific Literature:**

The method is built on flow matching and rectified flow generative models but integrates the SO(3) symmetries in the training process.

**Theoretical Claims:**

No

---

> ### Author Rebuttal · Authors · 2025-04-01
>
> *We thank the reviewer for acknowledging the novelty of integrating reflow and distillation method for accelerating the sampling of conformer generation model. Please see our response below to other questions and comments:*
>
> **Weakness**
>
> 1. We want to emphasize that the major motivation of this paper is to improve the efficiency of diffusion/flow-based models to the level of cheminformatics tools for conformer generation through algorithmic innovations. Given the need of ultra-large scale (~$10^8$-$10^9$ compounds) virtual screening, diffusion/flow conformer generation model would become practical with improved sampling efficiency. Bearing the motivation in mind, we chose to implement a compact 4.7M-parameter equivariant GNN model that achieves significant sampling speedup while maintaining good generation quality. Comparing to larger transformer-based baselines, the model's speedup in few-step sampling compensates for reduced generation quality: the 2-step $\mathrm{AvgFlow_Reflow}$ model outperforms 3-step MCF-B in precision metrics. It also achieves ~58% of the 5-step ETFlow with ~40x speedup in sampling. We believe that the current 4.7M model has fulfilled this motivation.
>
>    That being said, we agree with the reviewer that achieving the SOTA in generation quality is valuable. We have implemented a more scalable diffusion transformer (DiT) with pairwise biased attention, similar to AlphaFold3 [1] and Prote&iacute;na [2], for conformer generation. Due to the limited time, we trained a 52M-parameter DiT model with *AvgFlow* objective for only ~124k steps and benchmarked it on the Drugs test set (see results [here](https://anonymous.4open.science/r/confgen_icml25_rebuttal-8406/benchmark.png)). We have also compared the performance of the DiT model trained with *AvgFlow*, Kabsch, and Conditional OT (same experiment as Sec 4.1 of manuscript, see results [here](https://anonymous.4open.science/r/confgen_icml25_rebuttal-8406/dit_obj_comp.png)). The benchmark results of DiT demonstrates that:
>
>     - *AvgFlow* can be a better training objective than Conditional OT and Kabsch even with **non-equivariant model architecture**.
>
>     - A transformer-based architecture with more parameters trained with *AvgFlow* can achieve performance on-par with other SOTA models, even with a relatively limited number of training steps.
>
>    The performance of the DiT model is expected to improve further as it is trained for more steps. It can also be scaled up to similar size to MCF-L for better performance if resources allow. We are currently working on reflow fine-tuning the DiT model and we will share results once completed.
>
> 1. We agree with the reviewer that an OOD evaluation on the GEOM-XL dataset can examine the model's generalizability to large molecules. The benchmark results of the 4.7M $\mathrm{AvgFlow}$ model on GEOM-XL is attached [here](https://anonymous.4open.science/r/confgen_icml25_rebuttal-8406/geom_xl.png). In general, our model has slightly higher mean AMR but on-par median AMR for both recall and precision compared with SOTA model MCF and ETFlow. We also want to mention that the DiT model we recently trained with *AvgFlow* objective is expected to generalize better to larger molecules thanks to its scalable architecture. We will benchmark its performance on GEOM-XL as well.
>
> **Answer to the question**
>
> We understand the confusion of the reviewer about the derivation in Sec 3.1. To clarify:
> - For the conformer generation problem, we define orbits $\hat{x}$ as **low-energy conformers** of a given molecule. Therefore the integral $ \int d\hat{x}\ \hat{q}(\hat{x})$ in Eq. 2, representing the entire conformer ensemble, can be written as $\sum_{\hat x \in \mathcal{X}} \hat q(\hat x)$, where $\mathcal{X}$ is the set of conformers and $\hat q(\hat x)$ is the weight associated with each conformer (also elaborated by line 144-152, right column).
> - Your understanding is correct that for a given conformer $\hat{x}$, the rotated molecule $g \cdot \hat{x}$ has the same energy state.
> - The *AvgFlow* method proposed is capable of integrating over all conformers (orbits) **and** the SO(3) group. In practice, we only integrate over the SO(3) group during training and sample one conformer in each epoch to approximate the expectation of the conformer ensemble. (line 168-170, right column). Hence, the flow-matching objective proposed in this work is SO(3)-*Averaged Flow*.
>
> References:
>
> [1] Abramson et al. "Accurate structure prediction of biomolecular interactions with AlphaFold 3." *Nature* (2024)
>
> [2] Geffner et al. "Proteina: Scaling Flow-based Protein Structure Generative Models." *ICLR* (2025).

---

> > ### Comment · Reviewer_Adi6 · 2025-04-08
> >
> > I thank the authors for their response and the additional experiments they have provided. I think the new results on the GEOM dataset look better and closer to the baselines. However, the performance on the XL dataset is still poor, and the authors mentioned they want to update the results with the new Transformer architecture. Also, as the authors mentioned, their purpose is to improve sampling efficiency with good quality (given that they show better performance with a larger model), how does this affect sampling? Are there still some gains in sampling time?

---

> > > ### Author Response · Authors · 2025-04-09
> > >
> > > Thank you very much for the comment and acknowledging the performance improvement with the new DiT architecture. We have now updated the benchmark of AvgFlow$_\mathrm{DiT}$ on the OOD dataset GEOM-XL. The results are updated in [link](https://anonymous.4open.science/r/confgen_icml25_rebuttal-8406/geom_xl.png) and also summarized in the table below:
> > >
> > > **OOD generalization results on GEOM-XL. Unit is Å**
> > > | Method|AMR-R Mean↓| AMR-R Med↓| AMR-P Mean↓|AMR-P Med↓|
> > > |-------|------------|------------|------------|------------|
> > > | MCF-S (13M)|2.22|1.97|3.17|2.81|
> > > | MCF-B (64M)|2.01|1.70|3.03|2.64|
> > > | MCF-L (242M)|1.97|1.60|2.94|2.43|
> > > | ET-Flow (8.3M)|2.31|1.93|3.31|2.84|
> > > | AvgFlow$_{\mathrm{DiT}}$ (52M)|2.09|1.78|3.08|2.62|
> > >
> > > The AvgFlow$_\mathrm{DiT}$ achieves lower AMR than ET-Flow, which can be attributed to the scalable architecture. It also has very close performance to MCF-B. However, we have to emphasize that MCF models use 1000-step DDPM sampling while our model uses only 100-step ODE sampling.
> > >
> > > In terms of the sampling efficiency, we further benchmark the single-step inference wall time of AvgFlow$_\mathrm{DiT}$ and compared with the projected single-step wall time from corresponding papers:
> > >
> > > **Single step inference wall time. Unit is ms**
> > > | Method|Wall time|
> > > |-------|------------|
> > > | Tor. Diff.|25.6|
> > > | ET-Flow (8.3M)|21.2|
> > > | MCF-S (13M)|19.1|
> > > | MCF-B (64M)|34.0|
> > > | MCF-L (242M)|44.7|
> > > | AvgFlow$_{\mathrm{DiT}}$ (52M)|14.6|
> > >
> > > Thanks to the `jit` compilation of `JAX`, AvgFlow$_{\mathrm{DiT}}$ is still 24%-43% faster than other SOTA models for each step. Combing that with the extraordinary one-shot generation performance after distillation (see previous [response to Reviewer aynM](https://openreview.net/forum?id=6uPcJtMgWN&noteId=gSPr4Hu81O)), our model still demonstrates significant speed up in sampling time.
> > >
> > > We appreciate your constructive feedback and hope our extra benchmark results and comments clarify your concern regarding the manuscript.

---

### Official Review · Reviewer_aynM · 2025-03-12

**Overall Recommendation:** 3

**Summary:**

This paper presents SO(3)-Averaged Flow Matching and Reflow-based Distillation, a novel approach aimed at improving the computational efficiency of molecular conformer generation. By explicitly incorporating rotational symmetries into the flow-matching framework and refining the transport trajectories through Reflow and distillation, the authors significantly reduce the computational cost of both training and inference. The method achieves a substantial speedup compared to existing diffusion and flow-based models, demonstrating up to 20–50× faster sampling while maintaining competitive conformer quality. The approach is evaluated on standard molecular datasets (GEOM-QM9 and GEOM-Drugs), where it is shown to outperform prior methods in terms of sampling efficiency and convergence rate.

**Claims And Evidence:**

The paper presents a novel approach for molecular conformer generation by integrating SO(3)-Averaged Flow Matching and Reflow-based Distillation. The primary claim is that by explicitly incorporating rotational symmetries into the flow-matching framework, the proposed method improves computational efficiency and convergence speed without compromising conformer quality. The authors provide empirical evidence demonstrating that their method is 20-50× faster in sampling compared to existing diffusion-based and flow-based methods while maintaining competitive performance on molecular benchmarks.

A secondary claim is that Reflow and distillation significantly enhance inference efficiency, reducing the number of sampling steps required for high-quality conformer generation. This is supported by experimental results showing that Reflow enables high-quality sampling in as few as two ODE steps, while distillation further reduces this to a single-step generation process. The authors further claim that their model outperforms larger transformer-based models (such as MCF and ET-Flow) in terms of speed and parameter efficiency.

**Essential References Not Discussed:**

The paper situates itself within the broader context of flow-based molecular generative models but does not explicitly compare its approach to certain relevant recent works. For example, the DiSCO (Diffusion Schrödinger Bridge for Molecular Conformer Optimization) model, which employs Schrödinger bridges for molecular conformer refinement, shares conceptual similarities with the proposed approach in terms of leveraging probabilistic flow-based modeling. However, there is no direct experimental comparison with DiSCO or other related works in diffusion-based molecular generation. A direct comparison, particularly in terms of sampling efficiency, quality trade-offs, and robustness across different molecular types, would be valuable in further contextualizing the contributions of this work.

https://ojs.aaai.org/index.php/AAAI/article/view/29238

**Experimental Designs Or Analyses:**

The authors evaluate their method on GEOM-QM9 and GEOM-Drugs, two widely used benchmarks, ensuring that the results are comparable to existing methods. The study includes quantitative metrics such as Coverage (COV) to measure the diversity of generated conformers, Average Minimum RMSD (AMR) to assess structural accuracy, and computational efficiency metrics (e.g., function evaluations per sample and total sampling time).

A key strength of the experimental setup is the ablation study, which isolates the effects of SO(3)-Averaged Flow Matching, Reflow, and Distillation. The results demonstrate that each component contributes to improved efficiency while maintaining competitive conformer quality. Additionally, the study benchmarks against a range of baseline models, including flow-based (MCF, ET-Flow) and diffusion-based (Torsional Diffusion) approaches, providing a fair comparison.

**Methods And Evaluation Criteria:**

The proposed methods, SO(3)-Averaged Flow Matching and Reflow-based Distillation, are designed to improve the efficiency of molecular conformer generation by addressing rotational symmetries and reducing the number of function evaluations (NFE) required for high-quality sampling. SO(3)-Averaged Flow Matching eliminates the need for explicit rotational alignment, reducing computational overhead during training, while Reflow and distillation straighten transport trajectories, enabling one-step or few-step sampling.

The evaluation criteria are well-aligned with standard practices in molecular conformer generation. The model is assessed using Coverage (COV), which measures how well the generated conformers match the diversity of ground-truth conformers, and Average Minimum RMSD (AMR), which quantifies structural accuracy. Efficiency is evaluated through sampling speed (microseconds per molecule) and NFE, directly comparing the computational cost of the proposed approach to state-of-the-art methods.

The experiments are conducted on GEOM-QM9 and GEOM-Drugs, two widely used benchmarks in conformer generation research. GEOM-QM9 consists of small, well-characterized molecules, while GEOM-Drugs includes more complex molecular structures, providing a rigorous test of the method’s generalizability.

**Other Comments Or Suggestions:**

The paper would benefit from a clearer discussion on the practical impact of efficiency gains, particularly in real-world molecular modeling workflows.

Performance degradation with Reflow on GEOM-Drugs should be further analyzed, with potential strategies to mitigate quality loss.

Releasing the complete official code would improve reproducibility and adoption by the research community.

Thorough grammar check could improve clarity (e.g. Line 312 ("to rotationally aligning") → "to rotationally align")

**Other Strengths And Weaknesses:**

The paper introduces a computationally efficient approach for molecular conformer generation, leveraging SO(3)-Averaged Flow Matching to eliminate rotational alignment and Reflow-based Distillation to enable fast sampling. It demonstrates a significant 20–50× speedup while maintaining competitive conformer quality, making it highly relevant for large-scale molecular screening. Comprehensive empirical validation and ablation studies further support its effectiveness.

However, while the complexity is reduced, the extent of performance improvement beyond computational efficiency is unclear, and the study does not fully establish whether the method leads to better conformer accuracy or diversity compared to prior approaches. The necessity of extreme speedup in practical applications remains uncertain, as conformer generation is typically an offline task. Additionally, performance degradation with Reflow on larger datasets (GEOM-Drugs) raises concerns about robustness, suggesting potential trade-offs between speed and quality. The lack of direct comparisons with transformer-based generative models (e.g., GeoMol, Equiformer, DiSCO) further limits a full assessment of its advantages.

A more detailed analysis of performance beyond efficiency, direct comparisons with a broader set of generative models, and a discussion of real-world impact would strengthen the study’s contribution.

I will decide whether to maintain the final score based on the authors' response to these issues.

**Questions For Authors:**

How well does the method generalize to highly flexible molecules with multiple low-energy conformers?

Does SO(3)-Averaged Flow perform robustly across different molecular sizes and bond constraints?

How does the approach compare with transformer-based generative models for molecular structures?

Would further fine-tuning with reinforcement learning improve molecular generation accuracy?

Can this method be extended to larger-scale drug discovery pipelines without significant modifications?

**Relation To Broader Scientific Literature:**

The paper builds upon prior work in flow-matching, diffusion-based molecular generation, and optimal transport methods. It extends conditional flow-matching by integrating rotational symmetry considerations, reducing computational overhead and improving efficiency. The introduction of SO(3)-Averaged Flow Matching eliminates explicit rotational alignment, while Reflow-based Distillation enhances sampling efficiency by reducing the number of required function evaluations.

However, the paper does not explicitly compare its approach to recent developments in rectified flow, flow-straightening, and diffusion-bridge-based techniques for conformer generation. Notably, DiSCO (Diffusion Schrödinger Bridge for Molecular Conformer Optimization, AAAI 2024, https://ojs.aaai.org/index.php/AAAI/article/view/29238) proposes a Schrödinger bridge-based diffusion model for molecular conformer refinement, offering an alternative perspective on optimizing transport trajectories. Additionally, transformer-based and SE(3)-equivariant generative models (e.g., GeoMol, Equiformer) have demonstrated strong performance in similar tasks but are not addressed in this study.

A more detailed discussion of these methods, along with empirical comparisons where feasible, would better contextualize the contributions of SO(3)-Averaged Flow Matching and Reflow-based Distillation within the broader landscape of molecular generative modeling.

**Theoretical Claims:**

The paper does not present new theoretical results or formal proofs but instead focuses on methodological advancements and empirical validation. While SO(3)-Averaged Flow Matching is conceptually motivated as a variance-reduced training objective, and Reflow-based Distillation is inspired by rectified flow methods, these are implemented as practical improvements rather than rigorously derived theoretical contributions. The method's effectiveness is demonstrated through empirical results rather than formal guarantees.

---

> ### Author Rebuttal · Authors · 2025-04-01
>
> *We want to thank the reviewer for the comprehensive review. Please see below for responses:*
>
> **Essential references**
>
> The references suggested by the reviewer are indeed relevant to this paper. However, we want to point out that we have explicitly compared our model to GeoMol for both the QM9 and Drugs benchmarks. We have also discussed DiSCO as a related work in the field of molecular conformer optimization in Sec 2.1 (line 60-62). From our perspective, DiSCO is a great method to optimize molecular conformer, which is a fundamentally different application compared to generating conformer from scratch (noise). Therefore, we did not benchmark explicitly against DiSCO. As far as we know, Equiformer has not yet been used for conformer generation. It would be nice if the reviewer could point us to related literature.
>
> **Weaknesses**
>
> >Conformer generation in virtual screening
>
> We want to respectfully correct the reviewer's statement about that the conformer generation being an offline task during virtual screening. During ultra-large virtual screening campaign (~$10^8$-$10^9$ compounds), conformer generation is indeed an *online* task, rendering the acceleration crucial. Our method, which achieves significant speedup, makes diffusion/flow model more practical for the use in virtual screening.
>
> >Degradation of performance after reflow
>
> We want to argue that the goal of reflow is to improve the model's performance in few-steps sampling, which is critical for achieving speedup in generation. Comparing to the model before reflow, the model after reflow demonstrates significantly better performance in <5-step sampling. We also want to emphasize that the reflow technique is model architecture-agnostic and can be applied to other flow-based models to accelerate generation. That being said, we are planning on reducing possible performance degradation by excluding lower quality generated conformer ($X_1'$) from the reflow fine-tuning dataset. In that way, we expect to alleviate the problem of sampling error propagating to fine-tuning stage.
>
> **Responses to comments**
>
> With the increasing demand of billion-level virtual screening, the  efficiency gain would be critically impactful to the application of flow-based conformer generation model. We would add more discussion about the impact in addition to line 42-43 of right column. Please see the previous answer for performance degradation of reflow and proposed future solution. We will perform a thorough grammar check for the future version of manuscript. We will release the model upon publication.
>
> **Answers to questions**
>
> 1. Many molecules in the Drugs test set are with >100 ground truth conformers. Therefore the method's ability to generalize to flexible molecules is reflected well by the benchmark results in Table 2.
>
> 1. Theoretically, *AvgFlow* as an training objective should not be affected by molecular size and bonds because those features are not used in the close-form solution.
>
> 1. We want to clarify that the major contributions of this work, including *AvgFlow* and reflow/distillation techniques, are training schemes rather than new model architectures. We believe the proper way of showing the advantage of *AvgFlow* is by comparing same architecture trained with different flow-matching objective (as shown in Fig.1). Similarly, we show in Fig.3 the necessity of reflow/distillation. To strengthen this point, we have recently trained a diffusion transformer with pairwise biased attention with the *AvgFlow* which achieves on par performance MCF and ETFlow (please see details in the response to reviewer iSgP). This demonstrates that the improvements brought by *AvgFlow* are architecture-independent.
>
> 1. We believe that RL-based fine-tuning can help models generate better conformers (more consistently at lower energy state) but may not improve sampling efficiency, which is the primary motivation of this work.
>
> 1. Yes the model can be extended to larger-scale drug discovery pipeline as it achieves significant speedup for flow-based model in conformer generation. The generation quality of our 1-step distilled model has surpassed cheminformatic tools.

---

> > ### Comment · Reviewer_aynM · 2025-04-04
> >
> > Thank you for the author's response. However, it fell slightly short of my expectations. I will maintain my score, and I respect any final decision by the AC if the paper is not accepted, as it has contributions but would benefit from further validation.

---

> > > ### Author Response · Authors · 2025-04-07
> > >
> > > Thank you for your comment on our response. We respect and value your opinion. We would like to provide an update here for you and other reviewers with additional benchmark results (see table below and also updated in [link](https://anonymous.4open.science/r/confgen_icml25_rebuttal-8406/benchmark.png)). These results pertain to the new diffusion transformer with pairwise biased attention (DiT) model, trained using *AvgFlow* and fine-tuned through reflow/distillation.
> > >
> > > After extended training (~360k steps), AvgFlow$_{\mathrm{DiT}}$ achieved performance on par with both MCF and ET-Flow-SS in conformer generation without sampling step limit. Specifically, it outperformed MCF in Precision metrics and surpassed ET-Flow-SS in Recall metrics.
> > >
> > > For 2-step generation, our AvgFlow$_{\mathrm{DiT-Reflow}}$ outperformed all baselines in Coverage metrics while ranking second only to ET-Flow in Precision metrics.
> > >
> > > Most notably, our AvgFlow$_{\mathrm{DiT-Distill}}$ significantly outperformed all baselines by a wide margin in 1-step generation. We want to emphasize that it surpassed Tor. Diff. (20 steps) with one-shot generation, despite Tor. Diff. starting generation with RDKit-generated conformers. Furthermore, it outperformed MCF-S (1000 steps) across all Precision metrics and exceeded the performance of all MCF and ET-Flow (2-step) models in Coverage metrics. Overall, the training strategy combining AvgFlow with Reflow/Distillation enabled a scalable transformer-based architecture like DiT to achieve exceptional one-shot conformer generation quality and diversity.
> > >
> > > We believe that further scaling of the DiT model can lead to even better performance. Additionally, the reflow/distillation technique will be increasingly beneficial for reducing inference costs as larger models are developed. For future work, we plan to explore model scaling and address performance degradation after reflow by filtering the reflow dataset. Upon publication, we will release the implementation details of the 52M DiT model.
> > >
> > > **Quality of generated conformer ensembles for GEOM-DRUGS (δ=0.75Å) test set**
> > > *Coverage (COV) and Average Minimum RMSD (AMR) in both Recall and Precision settings. Values are presented in the format of Mean (Median) for each metric. AvgFlow steps are averages due to adaptive step size. Models are categorized into __No step limit__, __2-step__, and __1-step__ generations. _Bold_ and `inline` values represent the best and the runner-up model in each category, respectively.*
> > >
> > > | Method| Step| COV-R (%) ↑| AMR-R (Å) ↓| COV-P (%) ↑| AMR-P (Å) ↓|
> > > |-------|-----|------------|------------|------------|------------|
> > > | **No Step Limit**||||||
> > > | RDKit|-|38.4 (28.6)|1.058 (1.002)|40.9 (30.8)|0.995 (0.895)|
> > > | OMEGA|-|53.4 (54.6)|0.841 (0.762)|40.5 (33.3)|0.946 (0.854)|
> > > | GeoMol|-|44.6 (41.4)|0.875 (0.834)|43.0 (36.4)|0.928 (0.841)|
> > > | Tor. Diff.|20|72.7 (80.0)|0.582 (0.565)|55.2 (56.9)|0.778 (0.729)|
> > > | ET-Flow-SS (8.3M)|50|79.6 (84.6)|0.439 (0.406)|**75.2** (**81.7**)|**0.517** (**0.442**)|
> > > | MCF-S (13M)|1000|79.4 (87.5)|0.512 (0.492)|57.4 (57.6)|0.761 (0.715)|
> > > | MCF-B (64M)|1000|`84.0` (`91.5`)|`0.427` (0.402)|64.0 (66.2)|0.667 (0.605)|
> > > | MCF-L (242M)|1000|**84.7** (**92.2**)|**0.390** (**0.247**)|66.8 (71.3)|0.618 (0.530)|
> > > | AvgFlow (4.7M)|102*|76.8 (83.6)|0.523 (0.511)|60.6 (63.5)|0.706 (0.670)|
> > > | AvgFlow$_{\mathrm{DiT}}$ (52M)|100|82.0 (86.7)|0.428 (`0.401`)|`72.9` (`78.4`)|`0.566` (`0.506`)|
> > > | **2-Step Generation**| | | | | |
> > > | MCF-B (64M)|2|46.7 (42.4)|0.790 (0.791)|21.5 (13.2)|1.155 (1.160)|
> > > | MCF-L (242M)|2|54.2 (54.4)|0.752 (0.746)|25.7 (18.8)|1.119 (1.115)|
> > > | ET-Flow (8.3M)|2|73.2 (76.6)|0.577 (0.563)|**63.8** (**67.9**)|**0.681** (**0.643**)|
> > > | AvgFlow$_{\mathrm{Reflow}}$ (4.7M)|2|64.2 (67.7)|0.663 (0.661)|43.1 (38.9)|0.871 (0.853)|
> > > | AvgFlow$_{\mathrm{DiT-Reflow}}$ (52M)|2|**75.7** (**81.8**)|**0.545** (**0.533**)|`57.2` (`59.0`)|`0.748` (`0.705`)|
> > > | **1-Step Generation**| | | | | |
> > > | MCF-B (64M)|1|22.1 (6.9)|0.962 (0.967)|7.6 (1.5)|1.535 (1.541)|
> > > | MCF-L (242M)|1|27.2 (13.6)|0.932 (0.928)|8.9 (2.9)|1.511 (1.514)|
> > > | ET-Flow (8.3M)|1|27.6 (8.8)|0.996 (1.006)|25.7 (5.8)|0.939 (0.929)|
> > > | AvgFlow$_{\mathrm{Distill}}$ (4.7M)|1|`55.6` (`56.8`)|`0.739` (`0.734`)|`36.4` (`30.5`)|`0.912` (`0.888`)|
> > > | AvgFlow$_{\mathrm{DiT-Distill}}$ (52M)|1|**76.8** (**82.8**)|**0.548** (**0.541**)|**61.0** (**64.0**)|**0.720** (**0.675**)|

---

### Decision · Program_Chairs · 2025-05-01

**Decision:**

Accept (poster)

**Comment:**

The reviewers recognize that this paper introduces a novel and computationally efficient approach—SO(3)-Averaged Flow Matching combined with Reflow-based Distillation—for molecular conformer generation, significantly improving sampling efficiency by up to 20–50× compared to existing methods. They particularly appreciate the extensive empirical evaluations, including careful ablation studies on standard benchmarks (GEOM-QM9 and GEOM-Drugs), clearly demonstrating the value of each methodological component. Additionally, the scalability demonstrated by applying the proposed methods across different model architectures underscores its potential broad impact.